# Introducing the MESMER-M-TPv0.1.0 module: spatially explicit Earth system model emulation for monthly precipitation and temperature

**Sarah Schöngart**[1,2], **Lukas Gudmundsson**[3], **Mathias Hauser**[3], **Peter Pfleiderer**[2], **Quentin Lejeune**[2], **Shruti Nath**[2], **Sonia Isabelle Seneviratne**[3], **and Carl-Friedrich Schleussner** TS1 [1,2,a]

[1]IRIThesys, HU Berlin, Friedrichstrase 191, 10117 Berlin, Germany
[2]Climate Analytics, Ritterstrasse 3, 10969 Berlin, Germany
[3]Institute for Atmospheric and Climate Science, ETH Zürich, Universitätsstrasse 16, 8006 Zurich, Switzerland
[a]now at: International Institute for Applied Systems Analysis, Schloßplatz 1, 2361 Laxenburg, Austria TS2

**Correspondence:** Sarah Schöngart (sarah.schoengart@climateanalytics.org)

**Abstract.** Emulators of Earth system models (ESMs) are statistical models that approximate selected outputs of ESMs. Owing to their runtime efficiency, emulators are especially useful when large amounts of data are required, for example, for in-depth exploration of the emission space, for investigating high-impact low-probability events, or for estimating uncertainties and variability. This paper introduces an emulation framework that allows us to emulate gridded monthly mean precipitation fields using gridded monthly mean temperature fields as forcing. The emulator is designed as an extension of the Modular Earth System Model Emulator (MESMER) framework, and its core relies on the concepts of generalised linear models (GLMs). Precipitation at each (land) grid point and for each month is approximated as a multiplicative model with two factors. The first factor entails the temperature-driven precipitation response and is assumed to follow a gamma distribution with a logarithmic link function. The second factor is the residual variability in the precipitation field, which is assumed to be independent of temperature but may still possess spatial precipitation correlations. Therefore, the monthly residual field is decomposed into independent principal components and subsequently approximated and sampled using a kernel density estimation with a Gaussian kernel. The emulation framework is tested and validated using 24 ESMs from the sixth phase of the Coupled Model Intercomparison Project (CMIP6). For each ESM, we train on a single-ensemble member across sce-

narios and evaluate the emulator performance using simulations with historical and Shared Socioeconomic Pathways (SSP5-8.5) forcing. We show that the framework captures grid-point-specific precipitation characteristics, such as variability, trend, and temporal auto-correlations. In addition, we find that emulated spatial (cross-variable) characteristics are consistent with those of ESMs. The framework is also able to capture compound hot–dry and cold–wet extremes, although it systematically underestimates their occurrence probabilities. The emulation of spatially explicit coherent monthly temperature and precipitation time series is a major step towards a computationally efficient representation of impact-relevant variables of the climate system.

## 1 Introduction

Earth system models (ESMs) are process-based models built on physical equations that govern the dynamic and thermodynamic process of the Earth system (e.g. Schneider et al., 2017). Their physically based modelling approach makes ESMs invaluable for understanding and explaining the impacts of human activities on the global climate. At the same time, the modelling approach is computationally expensive – generating a single ESM simulation for the sixth phase of the Coupled Model Intercomparison Project (CMIP6) takes weeks to months to complete (e.g. Balaji et al., 2017). This

limits the number of times any ESM can be run. However, studying a broad variety of different emission scenarios, along with estimating associated uncertainties and sampling natural variability, traditionally requires running an ESM many times (Lehner et al., 2020; Maher et al., 2021).

Emulators of ESMs are runtime-efficient models that approximate specific outputs of an ESM using statistical methods. An emulator (in this paper, the term emulator always refers to ESM emulators) is trained to approximate relationships between a set of predictor variables and selected target variable(s) from existing ESM data, which can then be applied to new predictor data. The temporal and spatial properties of the emulated target variable(s) should ideally be statistically indistinguishable from those of the actual ESM output. Emulators typically focus on a small set of key target variables, which reduces dimensionality and saves computational time, as well as storage. This is a reasonable choice, as for many downstream applications of ESM data only a small set of climatic variables is of interest; for example, the Large Ensemble output of the Community Earth System Model (CESM-LE) consists of 1168 climatic variables of which 64 % are virtually never downloaded, while 14 % contribute to over 90 % of downloads (Edwards et al., 2019). Emulators can generate thousands of realisations of ESM-like data, thereby overcoming the limitations of having only a small number of ESM realisations. As such, ESMs and emulators are complementary.

A number of such emulation frameworks of varying complexity exist. Some frameworks aim to approximate the mean trend of a single variable (e.g. Tebaldi and Arblaster, 2014), and references therein); others also emulate variability as either a stationary (e.g. Link et al., 2019) or a non-stationary (e.g. Nychka et al., 2018) process. Recent approaches target the simultaneous emulation of multiple variables to also correctly mimic cross-variable covariance structures (e.g. Tebaldi et al., 2022; Edwards et al., 2019; Liu et al., 2023). Emulators can also target different spatial and temporal scales (e.g. yearly in Beusch et al., 2020, or monthly in Nath et al., 2022). Emulators often use global mean temperature (GMT) as a forcing variable (e.g. Quilcaille et al., 2022), and some use additional drivers, such as ocean heat uptake, land–sea temperature contrast, or time-shifted GMT (e.g. Herger et al., 2015; Beusch et al., 2022).

In this study, we focus on generating emulations of monthly gridded land precipitation from monthly gridded land temperatures, while aiming to conserve the covariance structure between the two variables. Temperature and precipitation are two of the most important climatic variables and are required as input variables for most impact models (Lange, 2019).

There are already emulators targeted at jointly emulating temperature and precipitation. For example, Tebaldi et al. (2022) built their emulator STITCHES using resampling methods. They pool together all available data from any scenario, re-arrange them using constraints on global mean temperature, and then "stitch" the data back together. This enables STITCHES to generate multivariate spatially resolved emulations. However, the quality of the emulator is constrained by the amount of available ESM training data and does not perform ideally when data are under-representative. Link et al. (2019) have extended their temperature emulator, fldgen1.0, to also model precipitation (fldgen2.0; Snyder et al., 2019). Their framework relies on capturing the signal's mean response using pattern scaling (Tebaldi and Arblaster, 2014, and references therein) and then adding a variability term. The variability term possesses the spatiotemporal (cross-)correlations and is generated by decomposing the original ESM signal into its principal components (PCs), applying a Fourier transformation to the PCs, applying random phase shifts, and then back-transforming. Fldgen2.0 has been developed and tested for yearly data and implicitly assumes stationarity in the variability in the temperature and precipitation. Recently, Liu et al. (2023) developed a precipitation emulator, PrEMU, that targets the emulation of monthly gridded precipitation starting from monthly gridded temperatures. Their approach is able to deterministically reconstruct 70 % of the variance in global land average precipitation. However, PrEMU does not offer to emulate the remaining variance, and cross-variable covariances have not been verified.

In this study, we present a novel approach that aims at fully emulating land precipitation fields at monthly resolution, given a time series field of land temperatures, while especially approximating the cross-variable covariance structures. We show that the emulation framework closely resembles ESM output and even captures monthly compound extremes. Our emulator, called MESMER-M-TP, serves as an additional module within the MESMER (Modular Earth System Model Emulator with spatially Resolved output) framework (Beusch et al., 2020). MESMER has originally been designed to approximate the grid point level annual mean temperatures changes as a function of global mean temperature change, while explicitly accounting for spatial and temporal variability (Beusch et al., 2020). This approach has since been extended to also represent selected extreme weather indicators (MESMER-X), and key-impact-relevant variables such as fire weather and soil moisture (Quilcaille et al., 2022, 2023). A temporal downscaling module to emulate monthly climate output has also been successfully implemented (MESMER-M) (Nath et al., 2022). We here provide a module that can be coupled to MESMER-M temperature output (or to output from other emulators of monthly local temperatures) to generate bivariate temperature and precipitation emulations. The core of the approach employs generalised linear models (GLMs) (Dobson and Barnett, 2018). Our framework is easily extendable to other variables that follow distributions within the exponential family, and it allows for non-stationary variance functions.

This study is structured as follows. First, the methodological emulation framework is introduced in Sect. 2. Second, we

describe how the suggested methodology is applied to the ESM data in Sect. 3. To this end, we introduce the dataset (Sect. 3.1), give an overview on how the methodology is applied to the data (Sect. 3.2), and describe the validation of the emulation framework (Sect. 3.3). Next, we present our results in Sect. 4. Section 4 contains exemplary emulation output and validation metrics. Last, we summarise and discuss findings in Sect. 5. In addition, this paper comes with an extensive Appendix. Appendix B displays additional spatiotemporal validation metrics and complements Sect. 4. In Appendix C, we explain how MESMER-M-TP is coupled to an emulator that generates temperature data, and we carry out validation and uncertainty estimations for the coupled emulation chain.

## 2 Emulator description

### 2.1 Notation

$T$ and $P$ denote the spatially explicit monthly temperature and precipitation fields. We introduce the subscripts $s$, $m$, and $y$, such that $p_{s,m,y}$ ($t_{s,m,y}$) is the precipitation (temperature) value at location $s$ for month $m$ and year $y$. We set $m = 1$ as January and $m = 12$ as December. $P$ (and $T$) can be expressed as a 2-dimensional matrix, with columns corresponding to spatial locations and rows referring to specific month–year combinations, as follows:

$$P = \begin{pmatrix} p_{s_1,1,y_1} & p_{s_2,1,y_1} & \cdots & p_{s_l,1,y_1} \\ p_{s_1,2,y_1} & p_{s_2,2,y_1} & \cdots & p_{s_l,2,y_1} \\ \vdots & \vdots & \ddots & \cdots \\ p_{s_1,12,y_1} & p_{s_2,12,y_1} & \cdots & p_{s_l,12,y_1} \\ p_{s_1,1,y_2} & p_{s_2,1,y_2} & \cdots & p_{s_l,1,y_2} \\ \vdots & \vdots & \ddots & \cdots \\ p_{s_1,12,y_k} & p_{s_2,12,y_k} & \cdots & p_{s_l,12,y_k} \end{pmatrix}, \quad (1)$$

where $l$ denotes the number of spatial locations and $k$ the number of years. The matrix has dimensions $\dim(P) = (12 \times k) \times l$. The precipitation time series at location $s$ is the column vector $\boldsymbol{P}_s = (p_{s,1,y_1}, p_{s,2,y_1}, \ldots, p_{s,12,y_1}, p_{s,1,y_2}, \ldots, p_{s,12,y_k})^T$, where the superscript $T$ refers to the transpose. We define the grid-point-specific and month-specific precipitation as the time series consisting of precipitation samples from the same month over different years, $P_{s,m} = (p_{s,m,y_1}, p_{s,m,y_2}, \ldots, p_{s,m,y_k})^T$, meaning $P_{s,m}$ contains every 12th entry of $P_s$. All definitions work analogously for monthly temperatures.

### 2.2 General approach

The goal of the emulator is to derive the monthly spatially explicit precipitation based on the monthly spatially explicit temperatures. In particular, the emulated precipitation data should be spatially and temporally consistent with the temperature data. To this end, we suggest a multiplicative framework that can be summarised as

$$P_{s,m} = \underbrace{f_{s,m}\left(\{T_{r,m}\}_{r \in S_{s,m}}\right)}_{1} \times \underbrace{\eta_{s,m}}_{2}, \quad (2)$$

where the emulated local precipitation at grid point $s$ and for month $m$, $P_{s,m}$, is defined by two terms, starting with (1) the deterministic temperature-driven precipitation response. We assume that a large fraction of $P_{s,m}$ can be constructed from local temperature information. Let $S_{s,m}$ be the set of spatial locations with temperature time series that may contain relevant information for reconstructing $P_{s,m}$. We then use $\{T_{r,m}\}_{r \in S_{s,m}}$ to build a 2-dimensional predictor matrix $\mathbf{X}_{s,m}$. We assume that $\mathbf{X}_{s,m}$ relates to $P_{s,m}$ via the response function $f_{s,m}$. Note that $f_{s,m}$ acts independently on each grid point and for each month (see Sect. 2.3.2). Next, it is defined (2) by a stochastic multivariate noise term, $\eta$. $\eta$ is used to approximate the fraction of the natural variability that cannot be reconstructed from temperature information alone and thus appears random in our modelling framework. We assume that the precipitation residuals still possess information across locations and months but are independent of temperature. $\eta$ is evaluated at grid point $s$ and for month $m$.

The suggested framework is equivalent to assuming an additive model for the logarithm of precipitation, which is a common choice when modelling precipitation (e.g. Snyder et al., 2019; Gudmundsson and Seneviratne, 2016; McCullagh, 2019).

### 2.3 Temperature-driven precipitation response

The aim of the temperature-driven precipitation response is to capture the fraction of the precipitation signal that is deterministically derivable from temperature data. We do not assume a causal relationship here. Rather, the motivation is to provide for a consistent multivariate extension. To this end, we assume that temperature is a good predictor of the general trend in the precipitation signal, as well as parts of the variability. In order to capture both contributions simultaneously, we rely on the framework of GLMs (e.g. Dobson and Barnett, 2018; McCullagh, 2019). A GLM is a generalisation of ordinary linear regression and is applicable to any dependent variable that follows a specific distribution within an exponential family relative to the predictor variable(s). The basic assumption is that the dependent variable is related to a linear combination of the independent variables via a link function.

### 2.3.1 A GLM for precipitation

To apply the GLM framework to precipitation, we assume that $P_{s,m}$ follows a gamma distribution with shape parameter $k_{s,m}$ and scale parameter $\Theta_{s,m}$ conditioned on a set of temperature predictors accounting for the local and global temperature conditions, $\mathbf{X}_{s,m}$. The predictor matrix $\mathbf{X}_{s,m}$ is de-

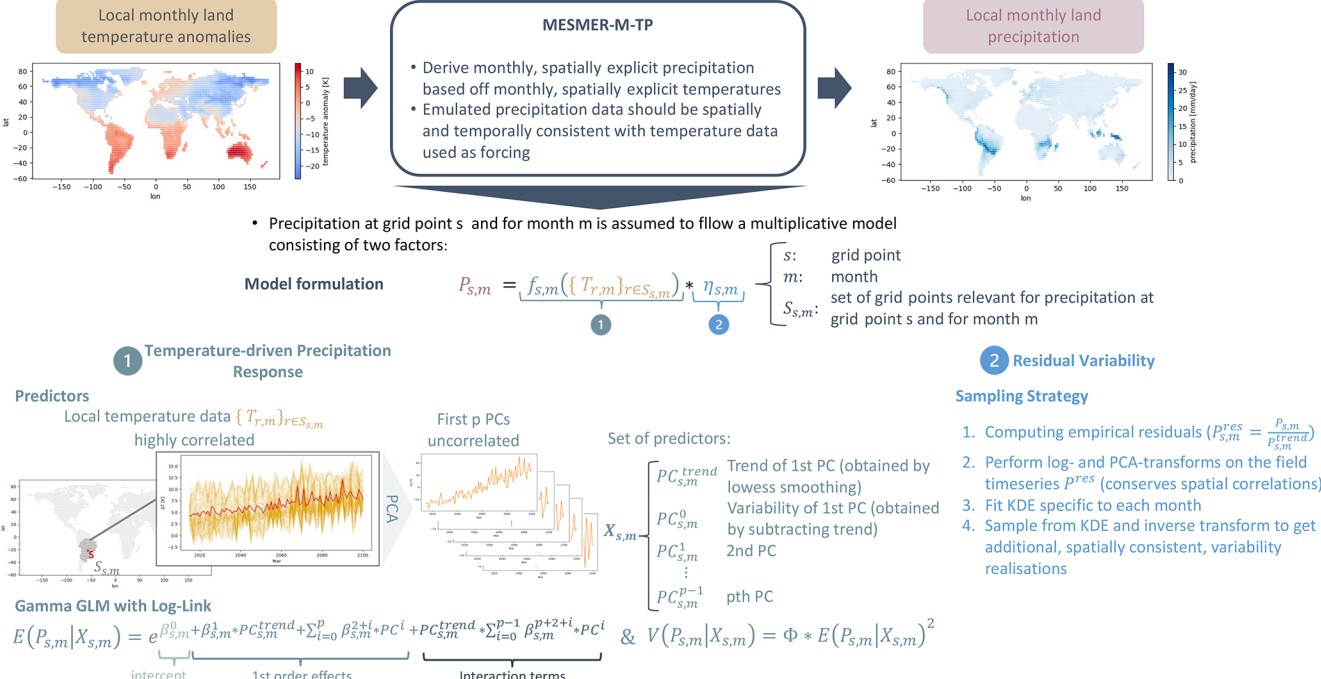

**Figure 1.** Schematic overview of our modelling approach. Precipitation is decomposed into a temperature-driven contribution and a contribution independent of temperature. We exploit local temperature information and the framework of gamma GLMs to reconstruct precipitation signals for each location and month. We then compute the empirical residuals, and after applying a log and a PCA transform to disentangle spatial correlations, we approximate the residuals individually using a KDE. The framework is described in more detail in Sect. 2.

rived from the set $\{T_{r,m}\}_{r \in S_{s,m}}$ of gridded temperature data, as described in Sect. 2.3.2. Precipitation is continuous and non-negative, while the gamma distribution is strictly positive. By replacing zero precipitation values with a small thresh-5 old for quasi-zero, the condition for the gamma distributions can be met. We choose a logarithmic link function, $g = \log$, such that the inverse link function is the exponential function, $g^{-1} = e$. Following this assumption, the response function $f_{s,m}$ is the expected value of $P_{s,m}$ conditioned on the 10 predictors $X_{s,m}$ (noted $E(P_{s,m}|\mathbf{X}_{s,m})$) as follows:

$$f_{s,m}\left(\{T_{r,m}\}_{r \in S_{s,m}}\right) = E(P_{s,m}|\mathbf{X}_{s,m})$$
$$= e^{\mathbf{X}_{s,m} \times \beta_{s,m} + \text{higher-order terms}}, \tag{3}$$

where $\beta_{s,m}$ is a vector of linear coefficients (see Sect. 3.2 for details on the higher-order terms). The mean value of a gamma distribution can also be expressed using its scale and 15 shape parameters as follows:

$$E(P_{s,m}|\mathbf{X}_{s,m}) = k_{s,m} \times \Theta_{s,m}. \tag{4}$$

Equally, we can express the variance of a gamma distribution as

$$V(P_{s,m}|\mathbf{X}_{s,m}) = k_{s,m} \times \Theta_{s,m}^2, \tag{5}$$

20 where $V(P_{s,m}|\mathbf{X}_{s,m})$ is the variance of $P_{s,m}$, conditional on the predictors $\mathbf{X}_{s,m}$ (noted $V(P_{s,m}|\mathbf{X}_{s,m})$). When fitting a

gamma GLM, $k_{s,m}$ is usually held constant, while the scale parameter $\Theta_{s,m}$ is varied. This leads to the mean–variance relationship of a gamma GLM,

$$V(P_{s,m}|\mathbf{X}_{s,m}) = \Phi_{s,m} \times E(P_{s,m}|\mathbf{X}_{s,m})^2, \tag{6}$$ 25

where $\Phi_{s,m}$ is called the dispersion and is given as the inverse of $k_{s,m}$ (therefore constant). As the conditional mean of the precipitation distribution is changing with the background climate, this variance–mean relationship offers to model non-stationary behaviour in the precipitation response. At the 30 same time, imposing the variance function of a gamma GLM is a strong assumption that may not hold true at all locations. Using a gamma GLM to model precipitation has been shown to yield good approximations in other studies (George et al., 2016; Hauser et al., 2017; Chandler, 2020; Kemsley 35 et al., 2024; Gudmundsson and Seneviratne, 2016), and in Sect. 3.3, we empirically validate our choice.

### 2.3.2 The predictor matrix $\mathbf{X}_{s,m}$

Precipitation is a complex climatic variable that depends on many factors such as water availability, temperatures, or 40 the terrain (Allen and Ingram, 2002; Trenberth et al., 2003; Tabari, 2020). As the goal is to reconstruct precipitation signals using temperature information only, we try to exploit local temperature information as much as possible. We assume that, in order to construct $P_{s,m}$, temperature informa- 45

tion at all grid points $r$ in proximity to $s$ is relevant. We denote $S_{s,m}$ as the set of $n$ spatial locations that are closest to $s$, and we assume that all $\{T_{r,m}\}_{r \in S_{s,m}}$ are relevant predictors for $P_{s,m}$. As the time series in $\{T_{r,m}\}_{r \in S_{s,m}}$ are highly correlated, we perform a principal component analysis (PCA) transform and only keep the first $p$ components. That is, we project $\{T_{r,m}\}_{r \in S_{s,m}}$ onto its $p$-dimensional eigenspace spanned by the vectors $\{\text{PCA}_{s,m}^i\}_{i \in \{0,...,p-1\}}$. The first principal component, $\text{PCA}_{s,m}^0$, is now expected to contain a strong trend. As precipitation may scale differently with temperature information on different timescales, we decompose $\text{PCA}_{s,m}^0$ into a trend and a variability term. The trend term is derived by locally weighted scatterplot smoothing (LOWESS), consistent with the methodology in Beusch et al. (2022). This leaves us with $\hat{T}_{s,m} = \{\text{PCA}_{s,m}^{0,\text{trend}}, \text{PCA}_{s,m}^{0,\text{var}}, \text{PCA}_{s,m}^1, \text{PCA}_{s,m}^2, ..., \text{PCA}_{s,m}^{p-1}\}$ as a set of feasible predictors. $\mathbf{X}_{s,m}$ is then constructed using the set $\hat{T}_{s,m}$ as row vectors and by adding column of ones to allow for a constant intercept, as follows:

$$\mathbf{X}_{s,m}$$
$$= \begin{pmatrix} 1 & \text{PCA}_{s,m,y_0}^{0,\text{trend}} & \text{PCA}_{s,m,y_0}^{0,\text{var}} & \text{PCA}_{s,m,y_0}^1 & \text{PCA}_{s,m,y_0}^2 & \cdots & \text{PCA}_{s,m,y_0}^{p-1} \\ 1 & \text{PCA}_{s,m,y_1}^{0,\text{trend}} & \text{PCA}_{s,m,y_1}^{0,\text{var}} & \text{PCA}_{s,m,y_1}^1 & \text{PCA}_{s,m,y_1}^2 & \cdots & \text{PCA}_{s,m,y_1}^{p-1} \\ \vdots & \vdots & \vdots & \vdots & \vdots & \ddots & \vdots \\ 1 & \text{PCA}_{s,m,y_k}^{0,\text{trend}} & \text{PCA}_{s,m,y_k}^{0,\text{var}} & \text{PCA}_{s,m,y_k}^1 & \text{PCA}_{s,m,y_k}^2 & \cdots & \text{PCA}_{s,m,y_k}^{p-1} \end{pmatrix}.$$
$$(7)$$

For simplicity, we sometimes refer to $\text{PCA}_{s,m}^{0,\text{trend}}$ as $\text{PCA}_{s,m}^{\text{trend}}$ and $\text{PCA}_{s,m}^{0,\text{var}}$ as $\text{PCA}_{s,m}^0$. The model design also offers to include higher-order effects by, for example, including $\mathbf{X}_{s,m}^2$ as a predictor or by including pairwise interactions between predictors. Including higher-order effects is a calibration choice and is discussed in Sect. 3.2.

## 2.4 Residual variability

We define the residual variability in the precipitation signal, $P_{s,m}^{\text{res}}$, as the fraction of precipitation that cannot be derived using gridded temperatures alone and assume it follows a multivariate stochastic process, $\eta_{m,s}$, as follows:

$$P_{s,m}^{\text{res}} = P_{s,m} / f_{s,m}(X_{s,m}^T) \propto \eta_{s,m}. \quad (8)$$

We assume that $P_{s,m}^{\text{res}}$ is independent of temperature but dependent on all other precipitation residuals. That is, the field $P^{\text{res}}$ still possesses spatial correlations, meaning we assume that grid cells in proximity to one another are likely very similar. As a gamma GLM does not contain any explicit assumptions about the distribution of the residuals, our goal is to generate new residuals with a distribution closely resembling the distribution of the empirical residuals. The empirical residuals are non-negative, and we first map them onto the entire space of real numbers by applying a logarithmic transformation. Next, we further apply a PCA to resolve the spatial dependencies across precipitation residuals. This allows

us to approximate the distributions of the PCA components individually, rather than modelling the joint distributions of the actual residuals. Let $\{\text{PC}_{i,m}^{\text{res}}\}_{i \in 1,...,q}$ be the first $q$ PCs of $P_m^{\text{res}}$. We assume that the probability density function (PDF) of each of the PCs can be modelled as a superposition of many Gaussian distributions with width $h_m$, as follows:

$$\text{PDF}(\text{PC}_{i,m,y}^{\text{res}}) = \frac{1}{k \times h_m} \sum_{j=1}^{k} \frac{1}{\sqrt{2\pi}} e^{-\frac{1}{2} \times (\frac{\text{PC}_{i,m,y}^{\text{res}} - \text{PC}_{i,m,y_j}^{\text{res}}}{h})^2}, \quad (9)$$

where $k$ denotes the number of sample years. In other words, we characterise the random process $\eta_{s,m}$ by applying a kernel density estimation (KDE) with a Gaussian kernel to the PCs of the empirical residuals. In order to generate additional, random, and spatially coherent variability realisations, we draw new samples from the KDE and inverse transform (first inverse PCA transformation and then inverse logarithmic transformation).

## 2.5 Model parameters

To summarise the above approach, we first construct the grid point and month-specific predictor matrix $\mathbf{X}_{s,m}$ using local temperature information. This offers two hyperparameters: (i) the number of the $n$-closest spatial locations that still influence precipitation at location $s$, and (ii) the number of PCA components, $p$, that should be kept as predictors. In addition, we can chose to include higher-order terms (for example, dependencies on $\mathbf{X}_{s,m}^2$ or interaction terms). The matrix $\mathbf{X}_{s,m}$ has $p + 2$ columns (the first PC is divided into a trend and a variability contribution, and we have a column of ones to allow for a constant offset), leaving us with $p + 2$ parameters for each grid point and month (the parameters are encompassed in the coefficient vector $\beta_{s,m}$). $\beta_{s,m}$ is fitted using the framework of a gamma GLM and a log-likelihood estimation. As the residuals of a gamma GLM do not have a predescribed functional form, we are approximating the residuals using a KDE that relies on another hyperparameter, the smoothing parameter or bandwidth, $h_m$.

## 3 Emulator application

### 3.1 Data

The emulator is trained on monthly mean temperature and monthly mean precipitation data from CMIP6 experiments (Eyring et al., 2016) of 24 different ESMs (see Table A1 in Appendix A). In this study, the term temperature refers to temperature anomalies relative to the period 1850–1900, while precipitation refers to absolute precipitation. The ESM data went through a centralised pre-processing that includes the interpolation to a common $2.5° \times 2.5°$ latitude–longitude grid and was obtained from the CMIP6 next-generation archive (Brunner et al., 2020). As variables are emulated over land only, grid cells with a land area coverage of less than

a third are filtered out, resulting in 2652 land grid points. Monthly precipitation data can contain zero values and in some cases very small negative numerical residuals. Therefore, for each ESM, a cut-off for quasi-zero is introduced by replacing zero and negative values with half of the smallest non-negative precipitation value found in the entire dataset. Data from five scenarios that represent combinations of Shared Socioeconomic Pathways (SSPs) and Representative Concentration Pathways (RCPs) are used, namely SSP1-1.9 (notation indicating the combination of SSP1 and RCP1.9), SPP1-2.6, SPP2-4.5, SPP3-7.0, and SSP5-8.5, and the historical simulations are considered (O'Neill et al., 2016). We refer to these SSP–RCP combinations as SSPs or scenarios. Not all 24 models provide temperature and precipitation data for each SSP (see Table A1 in Appendix A).

For each ESM, the emulator is trained independently, based on a single-ensemble member across all available SSPs. The historical simulation and the SSP5-8.5 scenario of the remaining ESM ensemble members are used for evaluating the emulator performance and are referred to as validation runs. When generating emulations from actual ESM data, we generate a single precipitation realisation for each available temperature field. Therefore, the number of emulations exactly equals the number of ESM runs. A special focus is put on the three models with the highest number of validation runs: ACCESS-ESM1-5 (Ziehn et al., 2019), CanESM5 (Swart et al., 2019), and MPI-ESM1-2-LR (Schupfner et al., 2021). These three models offer at least 30 ensemble members each, which allows us to compare ensemble statistics and, in particular, extreme event distributions. As an example, ACCESS-ESM1-5 has 40 ensemble members (see Table A1). We calibrate on ensemble member "r1i1p1f1" across scenarios to then generate 39 precipitation emulations across scenarios based on the gridded temperatures from the remaining 39 ensemble members.

MESMER-M-TP has been designed as a module that can be coupled to existing temperature emulators. To additionally evaluate the emulator performance and the propagation of uncertainties in this context, the trained emulator is coupled to emulated monthly temperatures of the historical simulation and the SSP5-8.5 scenario. The emulated temperature dataset was specifically generated for this study and is described in Appendix C1. We generate an ensemble of 100 temperature and precipitation realisations per model and scenario.

## 3.2 Calibration

The methodological framework described in Sect. 2 offers hyperparameters (see Sect. 2.5) for both the temperature-driven precipitation response module and the residual variability module. As part of the temperature-driven precipitation response, $P_{s,m}$, is reconstructed from information in the $n$-closest temperature time series, $\{T_{r,m}\}_{r \in S_{s,m}}$, with $|S_{s,m}| = n$. For simplicity and comparability, we assume that $n$ is con-

stant across models, months, and grid points. Therefore, $S_{s,m}$ only depends on the spatial location and reduces to $S_s$. The choice of $n$ is a trade-off between model complexity (for higher $n$, the PCA has more coefficients and takes longer to compute) and prioritising local modes of variability over large-scale/global relationships. We find that across months and models, the strongest correlations between the variability in temperature and the variability in precipitation occur in almost 80 % of the cases within the closest 150 grid points. Thus, we set $n = 150$, such that we can derive precipitation based on the 150 closest temperature locations. We have tested the approach for a variety of $n$ and find that, across grid points and months, results for $n \in [75, 400]$ are comparable, while introducing larger $n$ is too computationally intensive. We also tested using a single global decomposition by setting $n = 2652$, which leads to good results in some areas (e.g. North America) and performs poorly in other regions (e.g. Southeast Asia). As the set of $\{T_{r,m}\}_{r \in S_s}$ are highly correlated, we apply a PCA transformation prior to using them as independent variables for the GLM (see Sect. 2.3.2). The amount of explained variance in each PC decreases rapidly over the first five PCs and strives towards zero with an increasing component number. To include as much information as possible, while not inflating the model, we set $p = 8$. It is possible to include higher-order terms in the model, that is, to add $\mathbf{X}_{s,m}^2$ as a predictor or allow for interaction terms. We found that the model performance improved when we allow for first-order interaction terms between the trend in the first PC and all other PCs. The physical interpretation begins so that the relative importance of the PCs varies with the trend in local temperatures. Including additional terms had little effect on the model performance. Therefore, the calibrated model equation for the trend contribution to precipitation reads as follows: TS3

$$
\begin{aligned}
f_{s,m} = \; & e^{\beta_{s,m}^0} \qquad \text{intercept} \\
& \times e^{\beta_{s,m}^1 \times \mathrm{PCA}_{s,m}^{0,\text{trend}} + \beta_{s,m}^2 \times \mathrm{PCA}_{s,m}^{0,\text{var}} + \sum_{p=1}^{7} \beta_{s,m}^{p+2} \times \mathrm{PCA}_{s,m}^p} \qquad \text{first order} \\
& \times e^{\mathrm{PCA}_{s,m}^{0,\text{trend}} \times \left( \beta_{s,m}^{10} \times \mathrm{PCA}_{s,m}^{0,\text{trend}} + \beta_{s,m}^{11} \times \mathrm{PCA}_{s,m}^{0,\text{var}} + \sum_{p=1}^{7} \beta_{s,m}^{p+11} \times \mathrm{PCA}_{s,m}^p \right)} \\
& \text{interaction.}
\end{aligned}
$$

(10)

Last, we set the parameters of the residual variability module. We apply a PCA on the precipitation residuals in order to resolve spatial correlations and treat the PCs independently. We keep 98 % of the variability in the original residual signals. The bandwidth of the KDE was chosen via $k$-fold cross-validation and was mostly constant across months and models. To reduce computational complexity, we have set $h_m = 0.1$ as a global parameter.

## 3.3 Validation

The validation framework consists of two steps: (1) evaluating the emulator's performance when it emulates precipita-

tion based on actual ESM temperatures and (2) evaluating the model's performance when it emulates precipitation based on emulated temperatures. The first evaluation step captures the direct error in the emulation framework, while the second step also captures the propagation of uncertainties from one emulator to another. Results for the former are shown in Sect. 4, while results for the latter are shown in Appendix C. The evaluation procedure and result metrics are the same in both cases and described in the following.

The emulator is trained on one ensemble member across all available scenarios (see Sect. 3.1). Temperature data from all remaining ensemble members are used to generate emulated precipitation data for the first evaluation step (for the second evaluation step, we use the temperature dataset described in Appendix C1 as forcing). Both emulation datasets are assessed against actual ESM precipitation data from all remaining ensemble members for the historical period (1850–1950) and the projections from the high warming scenario SSP5-8.5 (2015–2100) independently. The time intervals and the scenario are chosen, such that the emulator's behaviour in a stable period with limited climate change and its behaviour under an extreme high-warming scenario can be equally analysed. As the three models ACCESS-ESM1-5, CanESM5, and MPI-ESM1-2-LR are the only models that offer a large number of additional ensemble members for evaluation (30+), we focus on validating the emulation approach using these three models and only schematically show results for all other models. In addition, we base our evaluation on the AR6 regions (Iturbide et al., 2020), with an emphasis on four regions that represent a diverse set of geographies and precipitation trends, namely southern central America (SCA), northern Europe (NEU), central Africa (CAF), and Southeast Asia (SEA) (see also Fig. A1). We validate the following properties:

1. *Inter-annual trend and variability in the precipitation.* We aim at verifying the emulated estimates of inter-annual trends, as well as of year-to-year variability, in $P_{s,m}$ across regions. To this end, ESM and emulated (EMU) data are aggregated by AR6 region. Next, all quantiles between the 1st and 99th quantile are computed in steps of one and compared against one another for both the historic and the future period. In addition, we compute quantile deviations for the 10th, median, and 90th quantile for each region (see Nath et al., 2022; Beusch et al., 2020). The gamma GLM is mainly responsible for correctly estimating the trend in $P_{s,m}$, while the residual variability module determines the variability in $P_{s,m}$. Therefore, the deviations allow us to draw conclusions on the performance of both models.

2. *Month-to-month relationships of precipitation.* The emulator was fitted for each month independently and only implicitly inherits the month-to-month relationships from the temperature data. Therefore, we verify the month-to-month relationships using lagged auto-

correlations. At each grid point and for each ensemble member, the correlation between the precipitation time series and a temporally shifted version of the same precipitation time series is computed. The correlation coefficient is computed for each ESM run and each EMU run individually and then averaged to obtain a single ESM/EMU value per grid point.

3. *Spatial precipitation structure.* The spatial structure in the precipitation signal is partially inherited form the spatial structure of the temperature field and partially explicitly enforced through the sampling strategy of the residuals. We verify that the joint use of the GLM and the KDE produces spatially coherent precipitation fields. To this end, we compute the month-specific cross-correlation matrix between precipitation time series at different grid points for each ensemble member. More precisely, for a given month and ensemble member, we compute the correlation between precipitation at any given grid point and precipitation at all other grid points. As we have 2652 grid points, this results in a correlation matrix of dimension (2652, 2652), whose entry $(i, j)$ describes the correlation between $P_{s_i,m}$ and $P_{s_j,m}$. We compute the ensemble mean of the correlation matrices and compare emulated estimates against ESM estimates.

4. *Spatial precipitation–temperature cross-variable correlations.* The GLM relies on exploiting local temperature information to reconstruct precipitation. Verifying that spatial cross-variable correlations are approximated well verifies the modelling approach and is important for downstream applications that rely on spatially and temporally consistent temperature and precipitation data. We verify the cross-variable statistics by computing the cross-correlation matrix between the precipitation and the temperature field for each month and for each ensemble member individually. Similar to point (3), for a given month, precipitation at any grid point is correlated with temperature at any other grid points. This results in a correlation matrix of dimension (2652, 2652), whose entry $(i, j)$ describes the correlation between $P_{s_i,m}$ and $T_{s_j,m}$.

5. *Compound temperature–precipitation extremes.* As the mechanistic processes that govern the occurrence of extremes are very different to the processes that determine long-term trends, verifying mean temperature–precipitation correlations alone is not enough to draw conclusions about the joint distributions of the tails. Therefore, we verify compound extremes individually. At each grid point, the 10th and 90th quantiles of temperature and precipitation are computed across ensemble members for ESM and EMU. To assess hot–dry extremes, we count the number of times a projection lies above the 90th temperature quantile and is simultane-

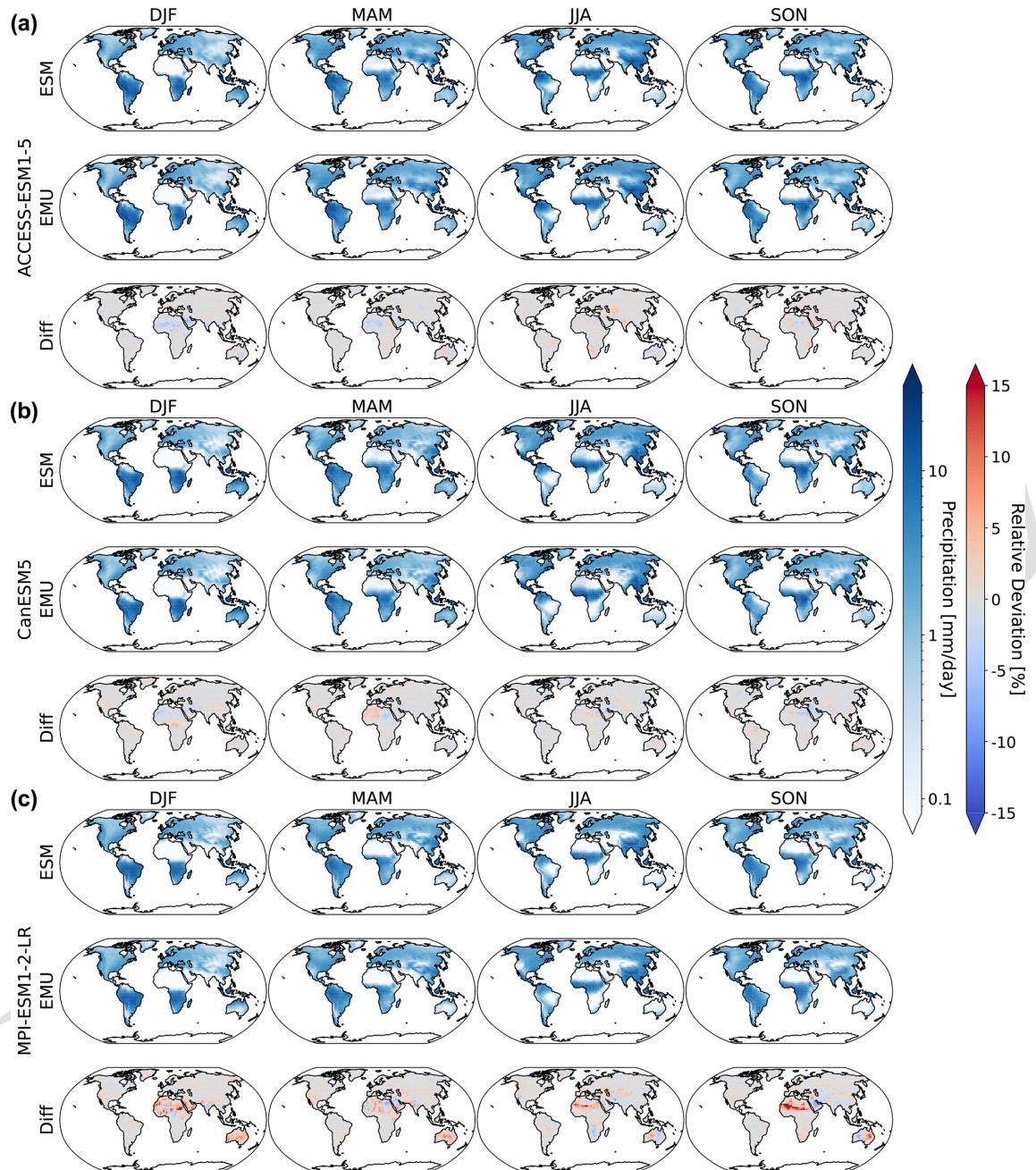

**Figure 2.** Maps of median gridded output aggregated by season under SSP5-8.5 for 2024. Panels **(a)**–**(c)** show results for ACCESS-ESM1-5, CanESM5, and MPI-ESM1-2-LR, respectively. From left to right, we show results for DJF, MAM, JJA, and SON. In each subfigure, the first row corresponds to simulations from the actual ESM ensemble, the second row corresponds to simulations from the emulated ensemble, and the third row shows the relative difference between actual and ESM data.

ously drier than the 10th precipitation quantile. Similarly, for cold–wet extremes, we count the number of times a projection is cooler than the 10th temperature quantile and simultaneously wetter than the 90th precipitation quantile. We then compute the mean across ESM and EMU estimates and scale the count to the number of events that would happen during the course of 100 years.

## 4  Results

This section is divided into two parts. In Sect. 4.1 we show all results that only concern precipitation characteristics;

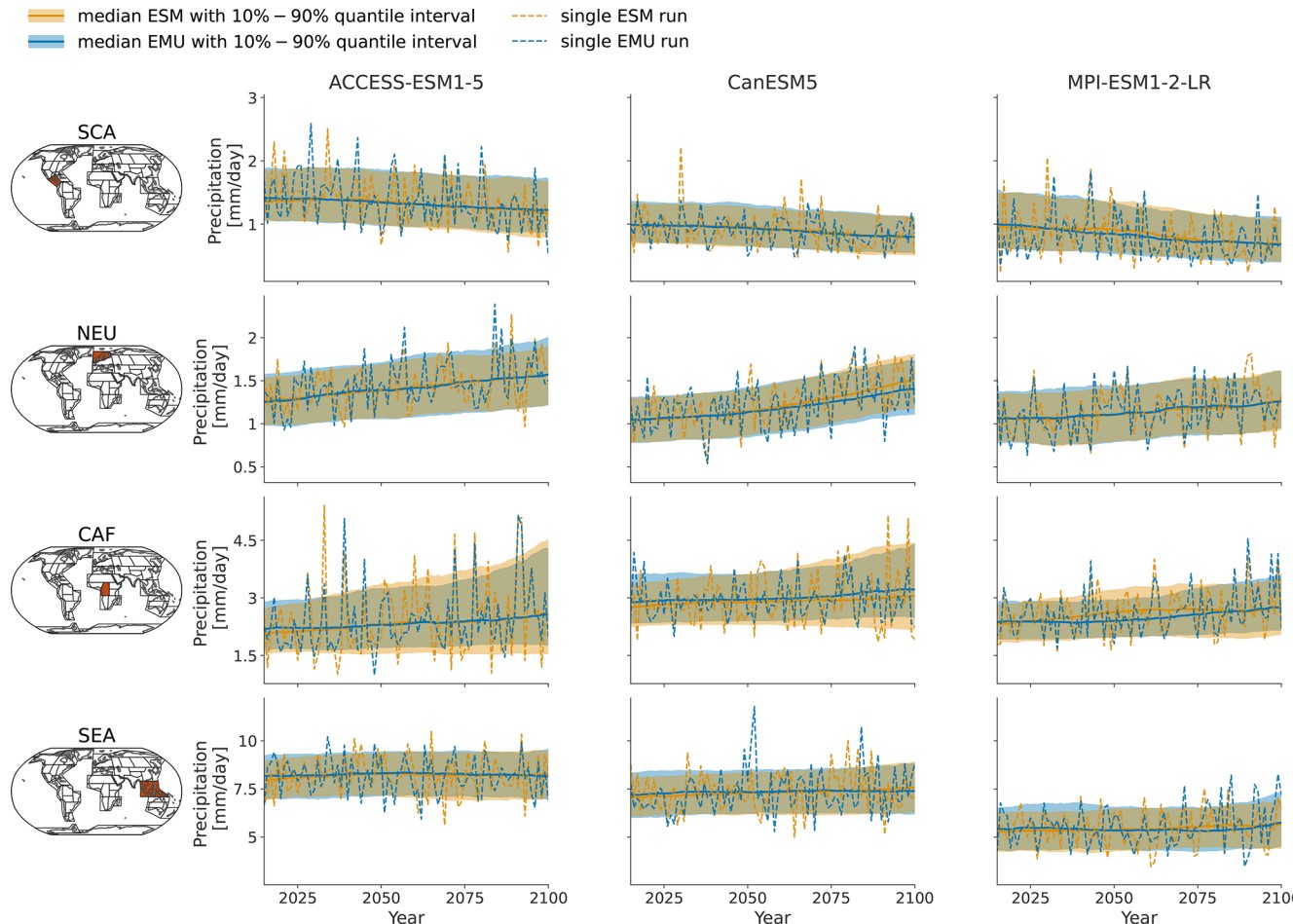

**Figure 3.** Time series of January precipitation from 2015–2100 for three different models (columns show ACCESS-ESM1-5, CanESM5, and MPI-ESM1-2-LR) averaged over four AR6 regions (rows show southern central America (SCA), northern Europe (NEU), central Africa (CAF), and South East Asia (SEA)). The time series highlight the year-to-year trend and variability in the January precipitation for SSP5-8.5. Orange (blue) line indicates the median ESM (EMU) time series, with shaded areas indicating 10 %–90 % quantile intervals. Dashed orange (blue) lines represent precipitation estimates of a single ESM (EMU) ensemble member. Note that the emulated time series was obtained using the ESM temperature field corresponding to the shown ESM precipitation time series as forcing.

these are inter-annual trend, inter-annual variability, month-to-month relationships, and the spatial structure of the precipitation signal (see properties (1)–(3) in Sect. 3.3). In Sect. 4.2, we show results concerning the joint characteristics of temperature and precipitation; these are the cross-correlation structures between temperature and precipitation, as well as compound extremes (see properties (4) and (5) in Sect. 3.3). We mainly focus on validating results when temperature fields from actual ESMs are used as forcing. In Appendix C, we show all results from this section when the emulator is forced with emulated temperatures.

### 4.1 Precipitation characteristics

In Fig. 2, we show exemplary maps of the median gridded precipitation ensemble output as seasonal averages over December–January–February (DJF), March–April–May (MAM), June–July–August (JJA), and September–October–November (SON) under SSP5-8.5 for 2024. The emulator adequately reproduces the precipitation patterns of the individual ESMs with the largest relative deviations occurring in dry regions (Sahara for MPI-ESM1-2-LR and ACCESS-ESM1-5; Australia for MPI-ESM1-LR). This is not surprising, as the relative error metric amplifies deviations when the divisor is close to zero. In addition, deviations are generally largest for MPI-ESM1-2-LR; this is equally expected given that, among the three models, the original MPI-ESM1-2-LR data have the smallest numerical representations of quasi-zero. However, even in the worst-performing cases, the relative deviations rarely exceed $\pm 15\,\%$, which pales in comparison to inter-model differences that reach deviations of more than $\pm 100\,\%$. The 10th and 90th quantiles of the gridded precipitation ensemble are displayed in

Figs. B1 and B1 and highlight similar deviations over dry regions.

As shown exemplarily in Fig. 3 for January, the emulator performs well in capturing inter-annual trends, as well as inter-annual variability, across months and models. The emulator captures different precipitation characteristics including quasi-stationarity (MPI-ESM1-2-LR in SEA), the shift in the mean precipitation (CanESM5 in SCA or ACCESS-ESM1-5 in NEU), and a widening of the distribution, resulting in an intensification of high-precipitation months (ACCESS-ESM1-5 in CAF) or an intensification of both high- and low-precipitation events (CanESM5 in CAF). As shown in Fig. B3, the emulator performs similarly, although slightly worse, for the month of July. In July, there are strong inter-model differences between precipitation projections from different models (SCA and CAF), suggesting low predictive accuracy in the models. In some cases (ACCESS-ESM1-5 in SEA and MPI-ESM1-2-LR in CAF), the emulator systematically overestimates high-precipitation events (90th percentile).

In Figs. 4 and B4, we explore these deviations from ESM quantiles in more detail. Systematic deviations only become apparent in the upper tail of the distribution (above 95th quantile in January and above 90th quantile in July), where the emulated values tend to lie above those from ESMs (MPI-ESM1-2-LR in SEA for January and all models in NEU for July). The emulated quantiles are usually within $\pm 10\%$ of the ESM quantiles. In particular, the deviations are small compared to inter-model differences (January uses SEA in MPI-ESM1-2-LR compared to SEA in ACCESS-ESM1-5). Our modelling framework implicitly assumes that non-stationarity in the variability in the precipitation can only be inherited from non-stationarity in the temperature signals through the gamma GLM. We do not account for potential non-stationarities in the residuals. The deviations in the tails of the distributions could indicate that this simplification is not strictly valid. We will discuss this in more detail in Sect. 5.

So far, we have only seen results for three models, for four regions, and for precipitation emulations based on actual ESM temperature data. Figure 5 gives an indication for the model performance in other regions, as well as for the difference in performance when emulating based on emulated temperatures (see Beusch et al., 2020; Nath et al., 2022). The coupled emulator (see Fig. B5; right panel) generally performs well in regions where the direct emulation error is small (for example, Aotearoa / New Zealand `CE1` (NZ) or central North America (CNA)) and usually suffers from stronger deviations whenever there already is a non-negligible error in the direct emulations (MPI-ESM1-2-LR in Greenland/Iceland (GIC) or western Africa (WAF)). In some cases, the coupled framework amplifies existing errors (ACCESS-ESM1-5 in northwestern South America (NWS) and northeastern South America (NES)) or introduces new errors (ACCESS-ESM1-5 in northern South America (NSA)). However, the performance is robust across forcing data and regions. We cannot estimate the direct emulation error for all 24 models due to a lack of available ESM data (computing quantile deviations requires a large ensemble of emulated data that cannot be generated if we do not have sufficient gridded temperature data from ESMs). Therefore, in Fig. 6, we show the quantile deviations from the coupled emulations for all available models (in this case, we also emulate large ensembles of gridded temperatures, leading to sufficient amounts of data). The results are comparable to the deviations found for the three focus models. The emulation framework tends to slightly overestimate the 10th quantile and the 90th quantile, while it underestimates the 50th quantile (the same holds true for July; see Fig. 6). The underestimations of the 50th quantile over the historical period and the simultaneous overestimation of the 10th quantile could suggest that our modelling procedure struggles to adequately capture the full complexity of the signal. It seems that our trend estimates are too low, and there is too little variability. Potentially, higher-order terms would be required to better represent the trend. In addition, there is some systematic overestimation of the 50th quantile in the July estimates (see Fig. B6), particularly in western Africa (WAF), central Africa (CAF), and the Arabian Peninsula (ARP).

We do not impose any constraints on month-to-month variations in the precipitation, thereby implicitly assuming that precipitation inherits the correct temporal properties from the temperature data. In Fig. 7, we explore this simplified assumption using lagged auto-correlations. In general, lagged auto-correlations are captured very well and strongly decrease with increasing time lag. The lag-1 correlations are slightly underestimated (in particular in MPI-ESM1-2-LR) but yield a consistent spatial pattern even without explicitly enforcing this structure. In particular, there is a high inter-model agreement on the temporal precipitation structure. We further verify temporal characteristics at the grid point level in Fig. B11.

The spatial precipitation structure is partially constructed from the spatial correlations in the temperature field through the GLM but mainly enforced by relying on the sampling strategy of the variability module (see Sect. 2.4). In Fig. 8, we see that pairwise precipitation relationships are captured well by the model and note an overall good agreement across models and months (see Fig. B7). In particular, no systematic bias (for example, a systematic over- or underestimation) is visible. This suggests that the residual variability module is well-suited to capture the spatial precipitation structures. We look into spatial characteristics for a selected number of grid points in Fig. B10.

## 4.2 Joint temperature–precipitation characteristics

As the precipitation emulations are built from local temperature data, we expect spatial cross-variable relationships between temperature and precipitation to be depicted well. In

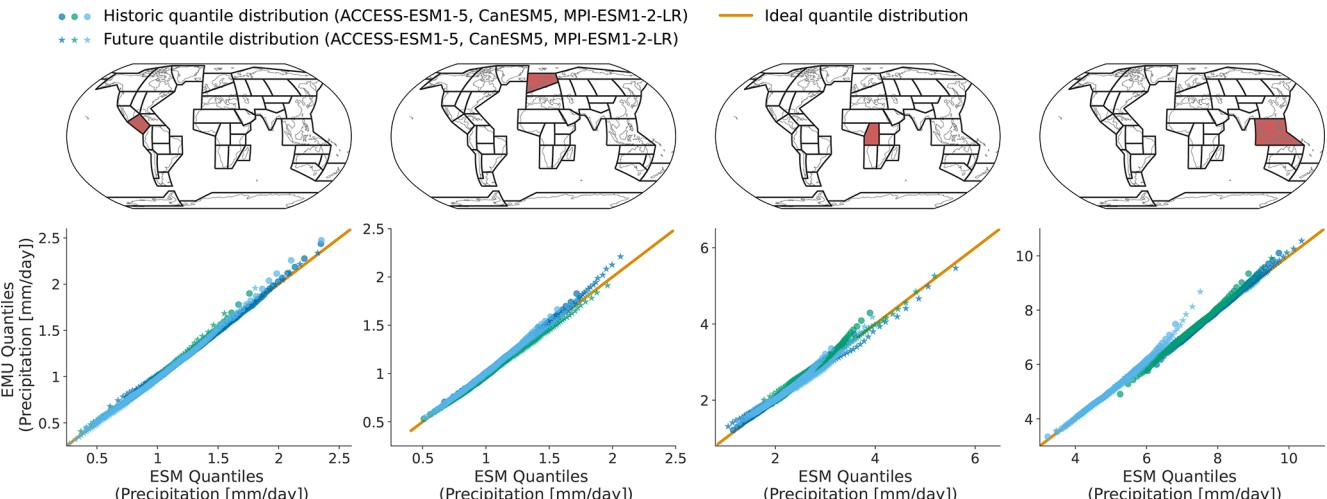

**Figure 4.** Quantiles (1st–99th in steps of one) derived from EMU data (*y* axis) scattered against estimates from ESM data (*x* axis) for four different regions (columns show SCA, NEU, CAF, and SEA). The quantiles were estimated for the historical period (1850–1950) and the future period (2015–2100) independently and are displayed individually (circles vs. stars). Colours are used to distinguish data from different models. Quantiles were derived as described in Sect. 3.3.

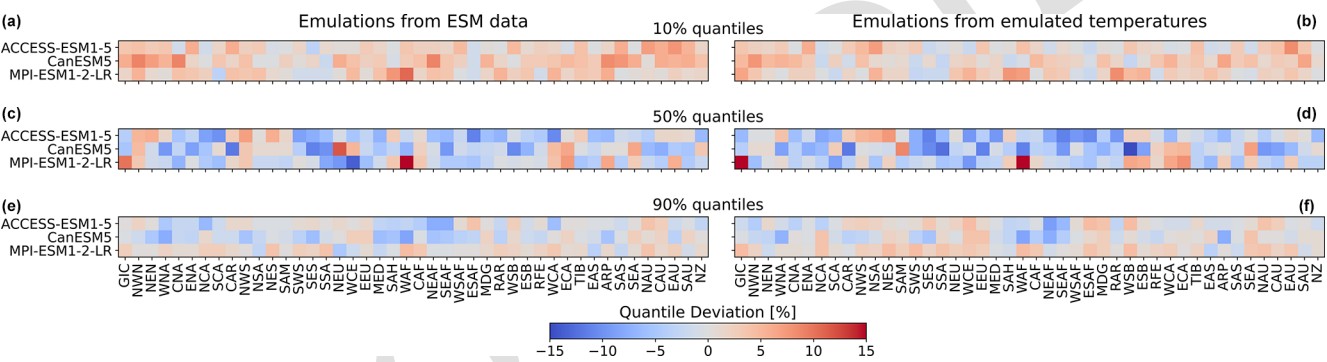

**Figure 5.** Regional deviations of January ESM precipitation from the 10 % **(a, b)**, 50 % **(c, d)**, and 90 % **(e, f)** quantiles of the emulations for the three focus models and across AR6 regions (see Fig. A1 for a map of the AR6 regions). **(a, c, e)** Emulations based on ESM temperatures. **(b, d, f)** Emulations based on emulated temperatures. Quantile deviations were computed over the historical period. Red (blue) indicates that the emulations project higher (lower) values compared to ESM data (see Sect. 3.3).

Fig. 9, we see that this is indeed the case. The emulator works particularly well if strong correlations are present (see also Fig. B8), while weaker correlations seem to be associated with larger errors. The strongest systematic errors (underestimation) occur for MPI-ESM1-2-LR. Even though precipitation is constructed from temperature signals in a certain proximity, strong long-range correlations are also approximated (see Fig. B8). Noteworthy is the strong intermodel disagreement in the strength and direction of the temperature–precipitation correlations for July (see Fig. B8); while CanESM5 projects fairly strong and positive long-range correlations, MPI-ESM1-2-LR projects moderate negative correlations.

Figure 10 displays the distribution of compound temperature and precipitation extremes. Our framework is generally able to capture compound temperature–precipitation ex-

tremes but typically underestimates them. In January, the strongest underestimations of both (hot–dry and cold–wet) extremes occur in Australia, central and southern Africa, and at the northeastern parts of South America. In July, the strongest underestimations are present over the Sahel region, the Arabian Peninsula, and the area adjacent to the Gulf of Mexico. The strength of the underestimation is comparable for January and July (see Fig. B9).

## 5 Discussion and conclusion

We have developed and validated an Earth system model (ESM) emulator that derives the monthly spatially explicit precipitation data from the monthly spatially explicit temperature data. We have shown that our framework captures temporal and spatial precipitation structures and produces realis-

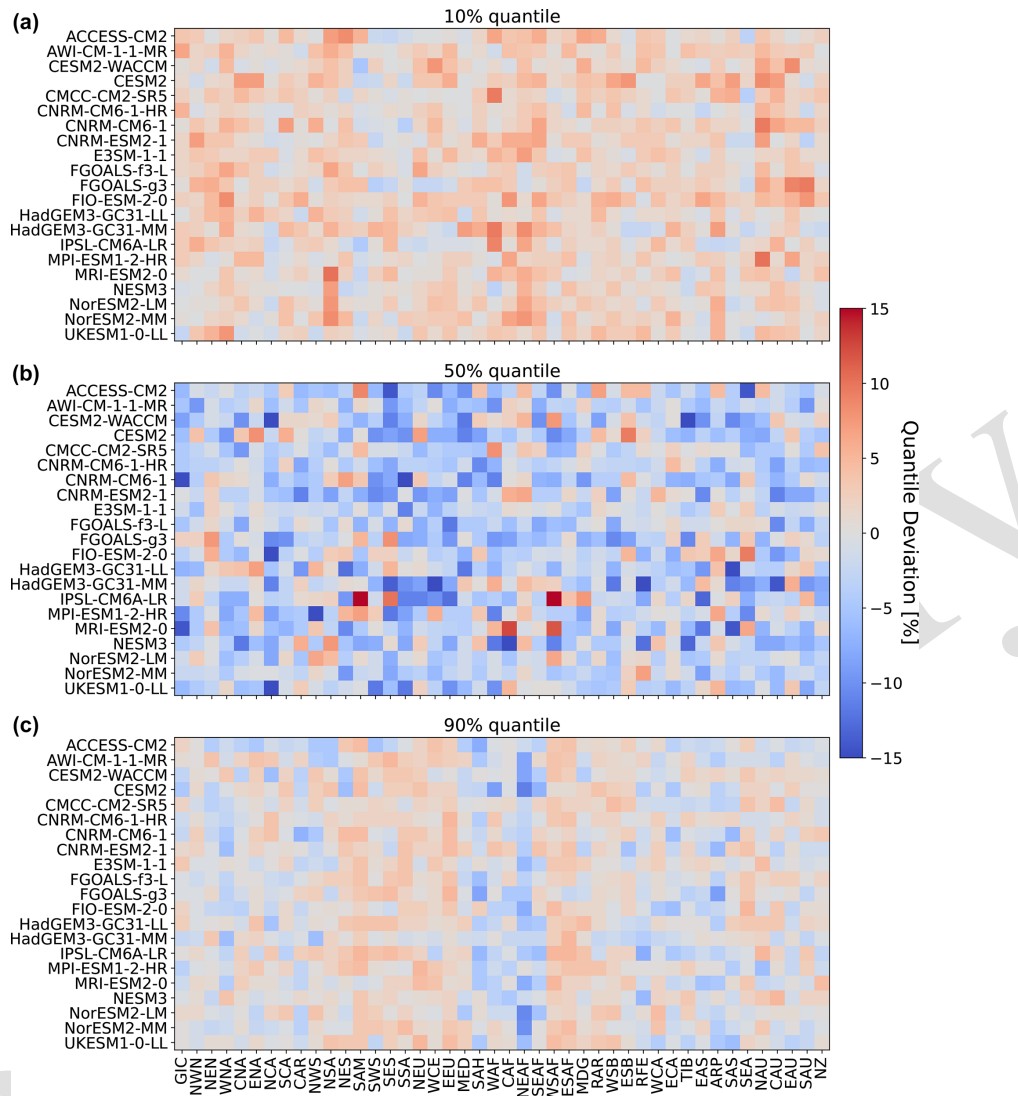

**Figure 6.** Same as the right panel in Fig. 5 but for the remaining 21 models.

tic cross-variable correlation structures. More precisely, we validated inter-annual trend and variability characteristics, along with month-to-month variability. The strongest deviations between the emulated and the ESM distributions occur in the tails of the precipitation distribution (mainly above the 95 % quantile), where we could observe some slight (usually smaller than 10 %) systematic overestimations of ESM quantiles. This might imply that some of the assumptions underlying the emulation framework do not hold anymore in extreme cases.

Extreme precipitation events can be driven by different physical processes and variables. For example, in low latitudes, very extreme precipitation events are often linked to the occurrence of tropical storms or cyclones (Khouakhi et al., 2017). The physical dynamics governing such singular events of strong convective precipitation are not resolved

in our statistical approach. We aim at modelling precipitation across different temporal scales and different spatial locations and relying on the same statistical model. This naturally comes with limitations. These limitations also become visible when jointly modelling temperature–precipitation extremes. Our framework is generally able to capture compound extremes and produces realistic spatial patterns. However, our emulator generally underestimates the occurrence of joint extremes. The emulator tends to slightly overestimate the magnitude of precipitation above the 95 % quantile, while simultaneously underestimating the occurrence rate of joint temperature–precipitation extremes, which suggests that the assumption of the precipitation residuals being independent of temperature is likely not fully accurate. In reality, the residuals are likely still not fully stationary and either depend on global or local temperature and potentially also the pre-

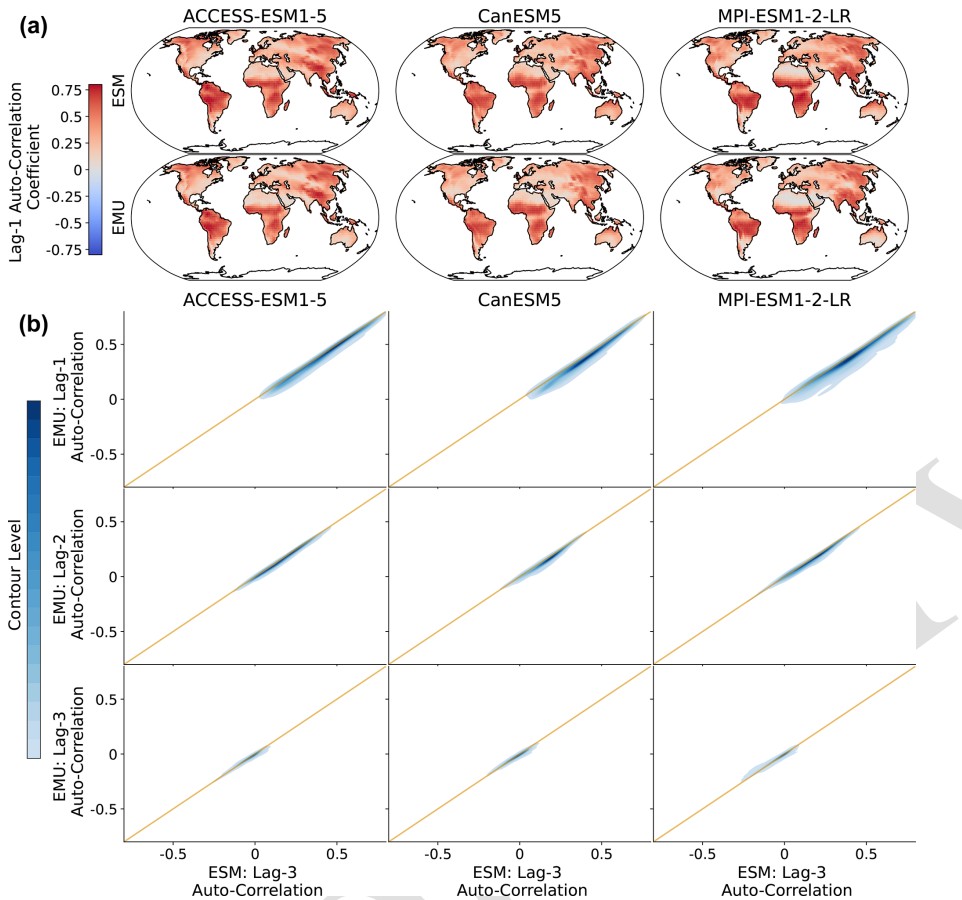

**Figure 7.** Lagged auto-correlations across grid points for three different models (columns). **(a)** Spatial distribution of lag-1 auto-correlations for ESM (EMU) data in the upper (lower) panel. **(b)** Distribution of emulated auto-correlations ($y$ axis) against ESM auto-correlations ($x$ axis) for three different time lags (lag-1, lag-2, and lag-3 in the upper, middle, and lower panels, respectively). The orange line represents the ideal distribution (EMU estimates exactly equal ESM estimates). The distribution was obtained using a KDE with contour levels in 5 % increments, such that every shade of blue represents 5 % of all data points, meaning 95 % of the data points lie within the coloured area, and 5 % lie outside of it.

dictions from the temperature-driven precipitation response. In addition, the emergence of compound extremes may depend on additional feedback effects, for example soil moisture in the case of heat–drought events (Jha et al., 2023). It is noteworthy, however, that the deviations of the emulated results from the actual ESM results are much smaller than inter-ESM differences.

In a next step, we have forced our precipitation emulator with emulated temperatures (see Appendix C1). The performance is comparable to the results obtained using ESM data as forcing (see Appendix C2).

There are multiple ways in which our approach could be further refined and adapted to different tasks. For once, instead of solely relying on gamma GLMs, thereby imposing a fixed mean–variance relationships at each location, the approach could be adjusted to optimise for other distribution families. In addition, the validation approach could be extended to other SSP scenarios – specifically to scenarios that

do not show continuous warming as transient – and quasi-equilibrium climate states have been show to have substantial local differences (King et al., 2021). To correctly model overshoot scenarios, it will also be necessary to include additional predictors. While local temperatures over land follow GMT to some extent under a reversal of the global mean temperature trend, changes in regional precipitation are not expected to be reversed in the short term in many regions (Pfleiderer et al., 2023). Beusch et al. (2022) have made some effort to overcome these difficulties by including ocean heat uptake as an additional predictive variable for local temperatures. Similar efforts could be pursued for precipitation. Last, the modelling framework could be improved by adjusting the residual variability module to account for a link between the predicted mean response and the distribution of additional variability. This would allow for non-stationary relationships in the variability module and would overcome some limitations in the tails of the distribution. We also note that our

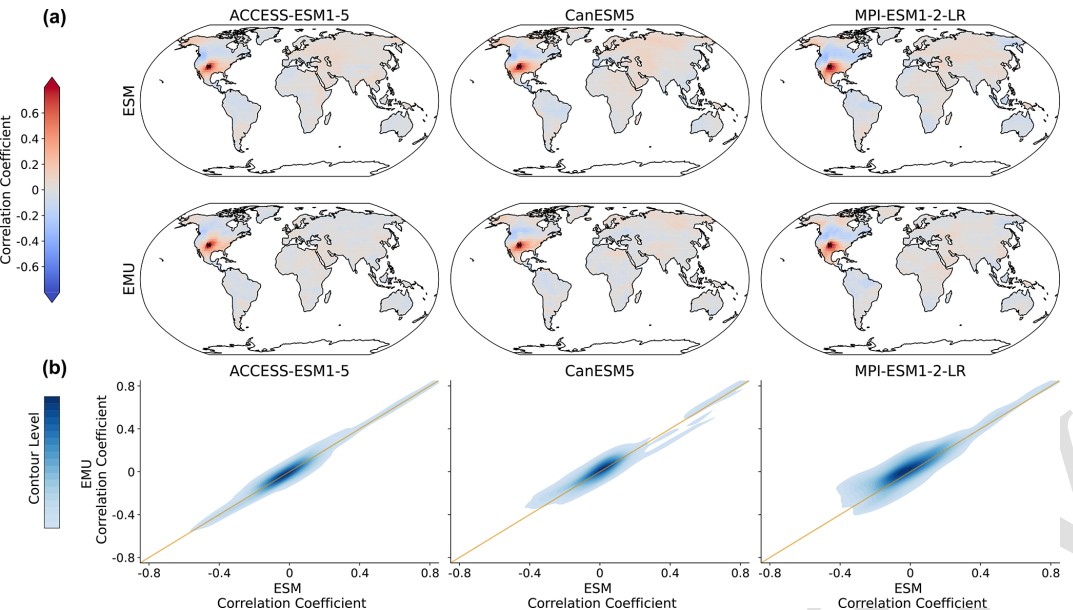

**Figure 8.** Spatial correlations between precipitation signals for January. **(a)** Correlation between precipitation at a randomly chosen grid point (New Mexico with coordinates $36.25°$ N, $-103.75°$ E; coloured in dark red, as the correlation of a time series with itself is 1) and precipitation at all other grid points for three models (columns show ACCESS-ESM1-5, CanESM5, and MPI-ESM1-2-LR) and for ESM (upper panel) and EMU data (lower panel). **(b)** Correlations between every possible combination of precipitation time series (that is, correlations between $P_{s,m=1}$ and $P_{r,m=1}$ for every possible combination of spatial locations $(s,r)$). EMU estimates ($y$ axis) are plotted against ESM estimates ($x$ axis). The orange line represents the ideal distribution (EMU estimates exactly equal ESM estimates). The distribution was obtained using a KDE with contour levels in 5 % increments, such that every shade of blue represent 5 % of all data points, meaning that 95 % of the data points lie within the coloured area, and 5 % lie outside of it.

model is not designed to resolve dynamics underlying long-range teleconnections such as those related, e.g., to the El Niño–Southern Oscillation. We see this as a promising area of future development.

To conclude, we offer a robust emulation framework for modelling spatially resolved monthly precipitation from spatially resolved monthly temperatures. In particular, the emulated precipitation field is spatially and temporally consistent with the temperature data used as forcing. Our emulation framework offers exciting new opportunities and is a step towards making climate science more accessible. While ESMs are costly and data-intensive to run, open-source emulators are available to everyone for projecting regional climate impacts. This is particularly important, as temperature and precipitation extremes are among the most impactful consequences of climate change. In addition, the emulator provides numerous applications, for example, coupling to impact models to provide an efficient modelling chain for translating emission scenarios directly into climate impacts. A promising avenue for this could be to couple our emulator to an emulator offering agricultural variables (e.g. Abramoff et al., 2023).

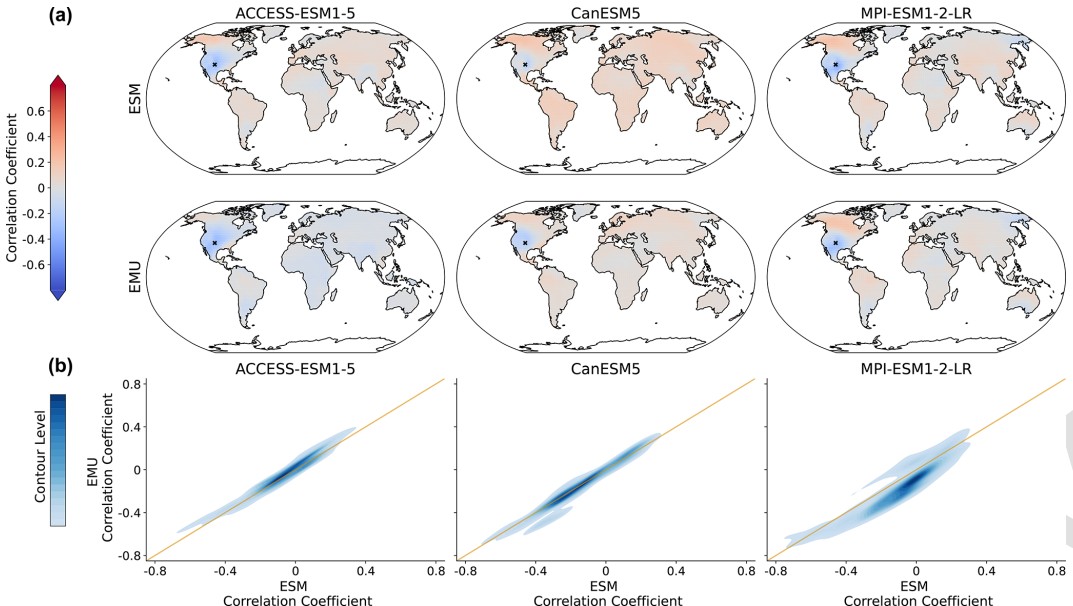

**Figure 9.** Spatial correlations between precipitation and temperature for January. **(a)** Correlation between precipitation at a randomly chosen grid point (New Mexico with lat 36.25 and long −103.75) and temperature at all other grid points for three models (columns of ACCESS-ESM1-5, CanESM5, and MPI-ESM1-2-LR) and for ESM (upper panel) and EMU data (lower panel). **(b)** Correlations between any possible combination of precipitation and temperature time series (that is, correlations between $P_{s_i,m=1}$ and $T_{s_j,m=1}$ for any possible combination of spatial locations $(s_i, s_j)$). EMU estimates ($y$ axis) are plotted against ESM estimates ($x$ axis). The orange line represents the ideal distribution (EMU estimates exactly equal ESM estimates). The distribution was obtained using a KDE with contour levels in 5 % increments, such that every shade of blue represents 5 % of all data points, meaning 95 % of the data points lie within the coloured area, and 5 % lie outside of it.

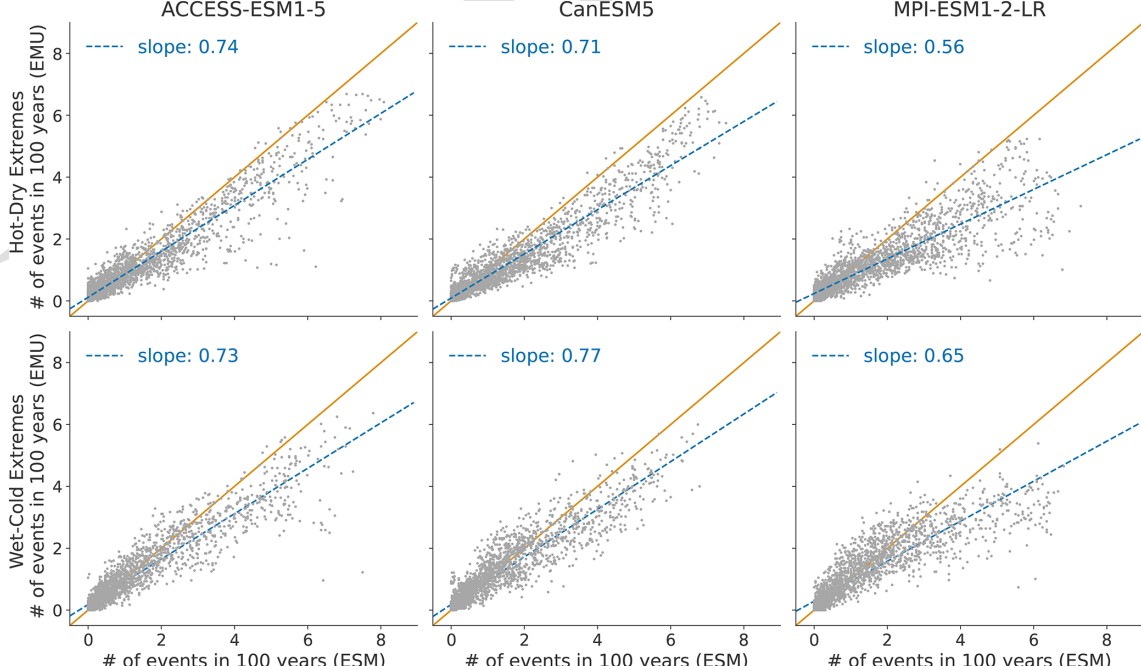

**Figure 10.** Distribution of compound temperature–precipitation extremes in January. Number of compound extreme events is estimated at each grid point (every dot represents one grid point) and counted, as explained in Sect. 3.3. Number of compound extremes found in EMU ($y$ axis) is plotted against the number of events found in ESM data ($x$ axis). Orange line represents the ideal distribution (number of events in EMU equals number of events in ESM).

## Appendix A: Earth system model data

**Table A1.** Overview of the 24 ESMs that are part of this study and the scenarios that are available for each model. The number of realisations includes only ensemble members that have data for all indicated scenarios. The training run column contains the identifier of the run that is used for training. All available runs, except the training run, are used for testing. If no testing run is available, we include the training run. The bold notation is used to indicate that we focused on the model during evaluation, as it has many ensemble members.

| Model name | Reference | Available scenarios | No. of realisations | Training run |
|---|---|---|---|---|
| ACCESS-CM2 | Dix et al. (2019) | SSP1-2.6, SPP2-4.5, SPP3-7.0, SPP5-8.5 | 5 | r1i1p1f1 |
| **ACCESS-ESM1-5** | Ziehn et al. (2019) | SSP1-2.6, SPP2-4.5, SPP3-7.0, SPP5-8.5 | **40** | r1i1p1f1 |
| AWI-CM-1-1-MR | Semmler et al. (2019) | SSP1-2.6, SPP2-4.5, SPP3-7.0, SPP5-8.5 | 1 | r1i1p1f1 |
| CESM2-WACCM | Danabasoglu (2019b) | SSP1-2.6, SPP2-4.5, SPP3-7.0, SPP5-8.5 | 3 | r1i1p1f1 |
| CESM2 | Danabasoglu (2019a) | SSP1-2.6, SPP2-4.5, SPP3-7.0, SPP5-8.5 | 1 | r1i1p1f1 |
| CMCC-CM2-SR5 | Lovato and Peano (2020) | SSP1-2.6, SPP2-4.5, SPP3-7.0, SPP5-8.5 | 1 | r1i1p1f1 |
| CNRM-CM6-1-HR | Voldoire (2019b) | SSP1-2.6, SPP2-4.5, SPP3-7.0, SPP5-8.5 | 1 | r1i1p1f2 |
| CNRM-CM6-1 | Voldoire (2019a) | SSP1-2.6, SPP2-4.5, SPP3-7.0, SPP5-8.5 | 6 | r1i1p1f2 |
| CNRM-ESM2-1 | Seferian (2019) | SSP1-1.9, SSP1-2.6, SPP2-4.5, SPP3-7.0, SPP5-8.5 | 5 | r1i1p1f1 |
| **CanESM5** | Swart et al. (2019) | SSP1-1.9, SSP1-2.6, SPP2-4.5, SPP3-7.0, SPP5-8.5 | **50** | r1i1p1f1 |
| E3SM-1-1 | Bader et al. (2020) | SPP5-8.5 | 1 | r1i1p1f1 |
| FGOALS-f3-L | Yu (2019) | SSP1-2.6, SPP2-4.5, SPP3-7.0, SPP5-8.5 | 1 | r1i1p1f1 |
| FGOALS-g3 | Li (2019) | SSP1-1.9, SSP1-2.6, SPP2-4.5, SPP3-7.0, SPP5-8.5 | 4 | r1i1p1f1 |
| FIO-ESM-2-0 | Song et al. (2019) | SSP1-2.6, SPP2-4.5, SPP3-7.0, SPP5-8.5 | 3 | r1i1p1f1 |
| HadGEM3-GC31-LL | Good (2019) | SSP1-2.6, SPP2-4.5, SPP3-7.0, SPP5-8.5 | 4 | r1i1p1f3 |
| HadGEM3-GC31-MM | Jackson (2020) | SSP1-2.6, SPP5-8.5 | 4 | r1i1p1f3 |
| IPSL-CM6A-LR | Boucher et al. (2019) | SSP1-1.9, SSP1-2.6, SPP2-4.5, SPP3-7.0, SPP5-8.5 | 7 | r1i1p1f1 |
| MPI-ESM1-2-HR | Schupfner et al. (2019) | SSP1-2.6, SPP2-4.5, SPP3-7.0, SPP5-8.5 | 2 | r1i1p1f1 |
| **MPI-ESM1-2-LR** | Schupfner et al. (2021) | SSP1-1.9, SSP1-2.6, SPP2-4.5, SPP3-7.0, SPP5-8.5 | **30** | r1i1p1f1 |
| MRI-ESM2-0 | Yukimoto et al. (2019) | SSP1-1.9, SSP1-2.6, SPP2-4.5, SPP3-7.0, SPP5-8.5 | 6 | r1i1p1f1 |
| NESM3 | Cao (2019) | SSP1-2.6, SPP2-4.5, SPP5-8.5 | 2 | r1i1p1f1 |
| NorESM2-LM | Seland et al. (2019) | SSP1-2.6, SPP2-4.5, SPP3-7.0, SPP5-8.5 | 1 | r1i1p1f1 |
| NorESM2-MM | Bentsen et al. (2019) | SSP1-2.6, SPP2-4.5, SPP3-7.0, SPP5-8.5 | 1 | r1i1p1f1 |
| UKESM1-0-LL | Good et al. (2019) | SSP1-2.6, SPP2-4.5, SPP3-7.0, SPP5-8.5 | 5 | r1i1p1f2 |

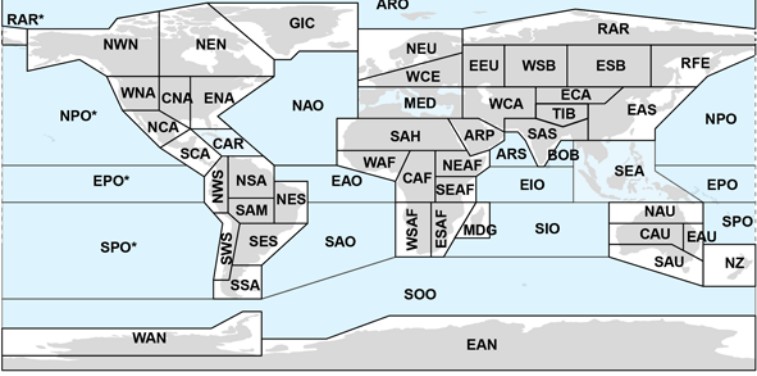

**Figure A1.** Map of all AR6 regions, including the 46 regions over land. Source: Iturbide et al. (2020).

## Appendix B: Additional validation for emulations from ESM data

This section is complementary to Sect. 4 and provides all additional validation metrics.

### B1  Seasonal validation

This section shows additional seasonal gridded precipitation output.

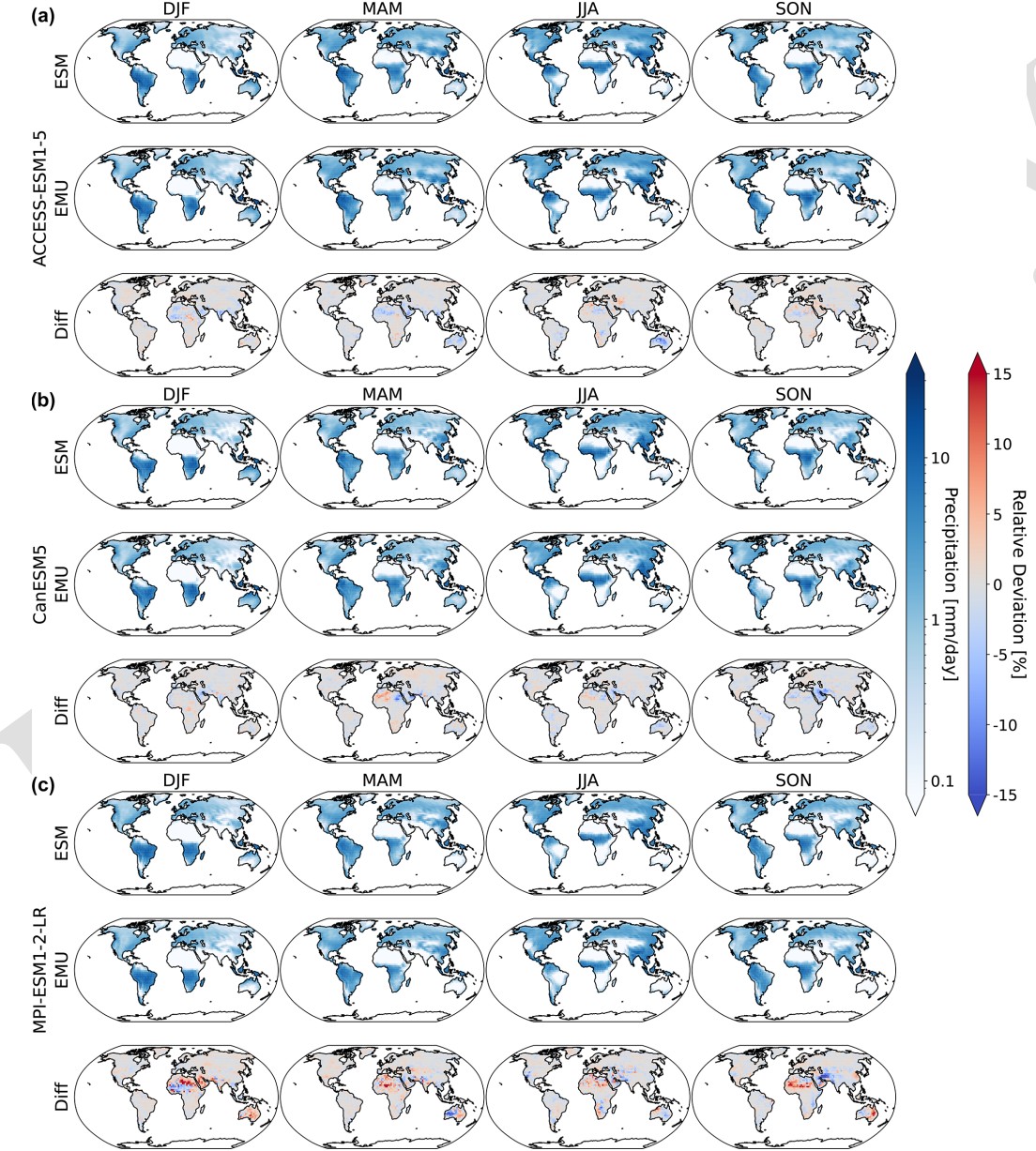

**Figure B1.** Same as Fig. 2 but for the 10th quantile.

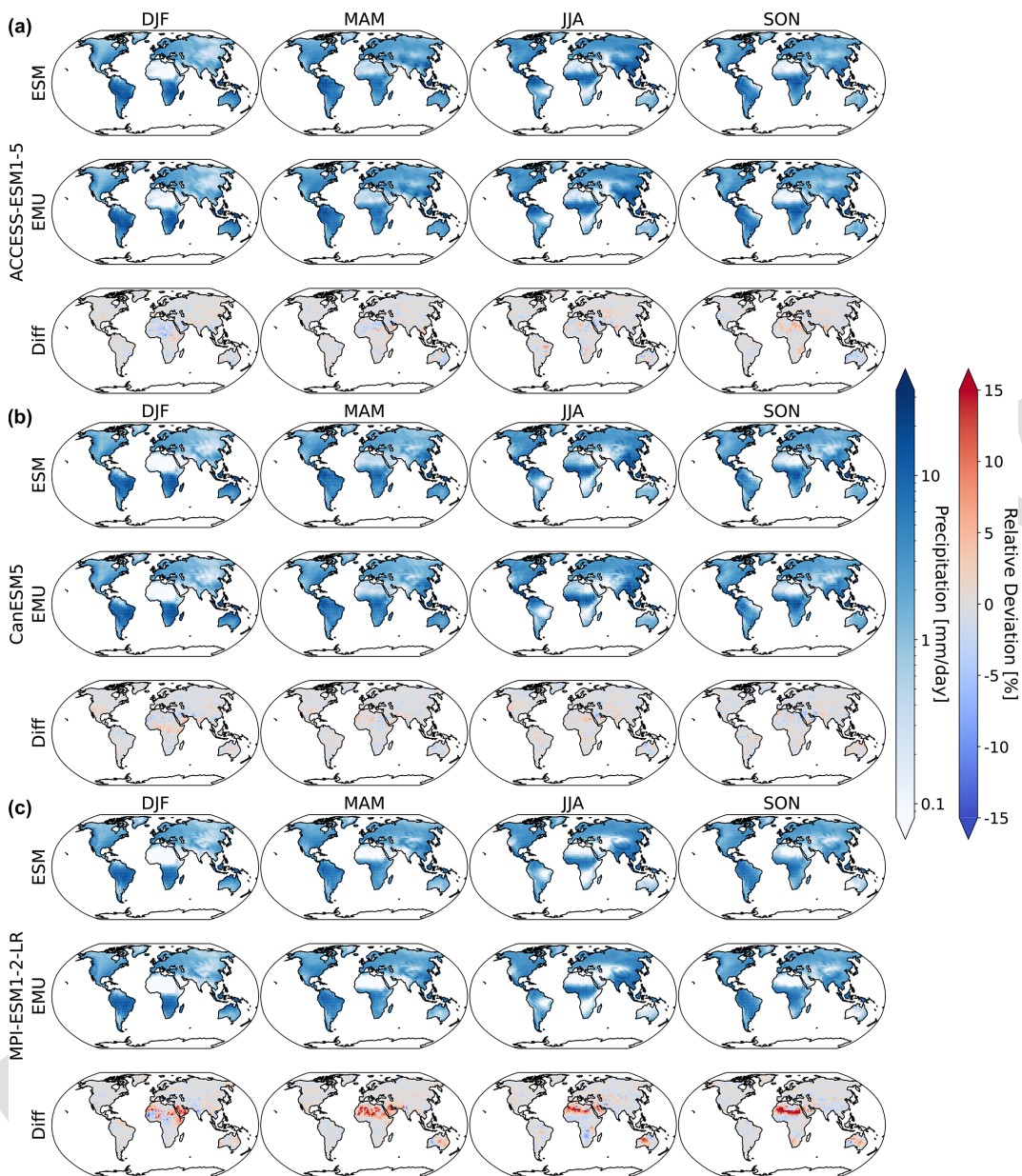

**Figure B2.** Same as Fig. 2 but for the 90th quantile.

## B2   Validation July

This section includes all graphics that were displayed for the direct emulation error in Sect. 4 but for July.

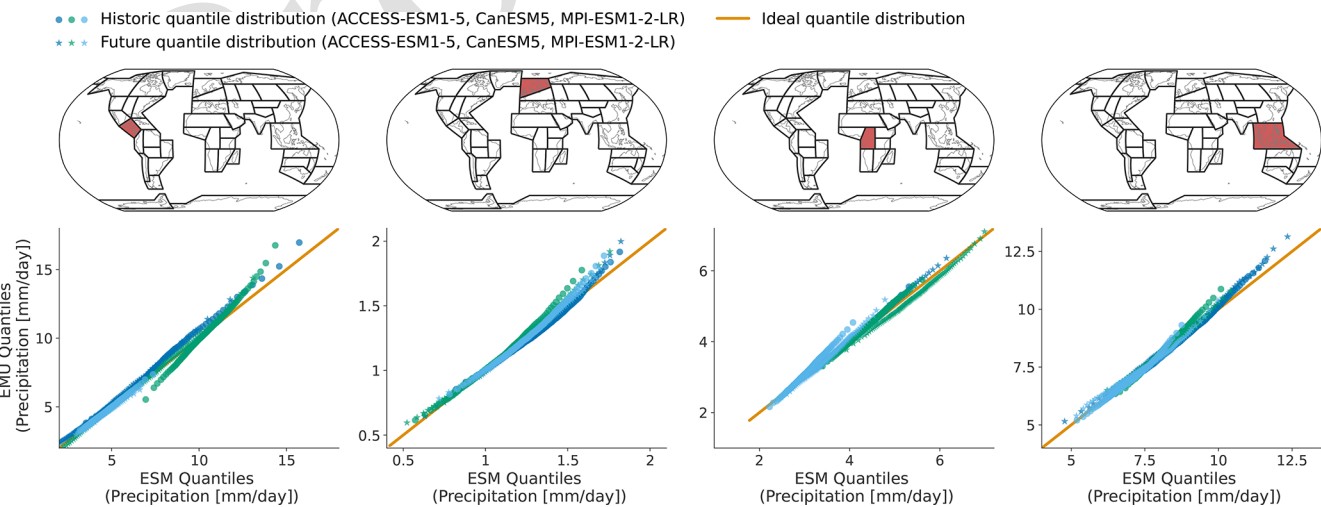

**Figure B3.** Same as Fig. 3 but for July.

**Figure B4.** Same as Fig. 4 but for July.

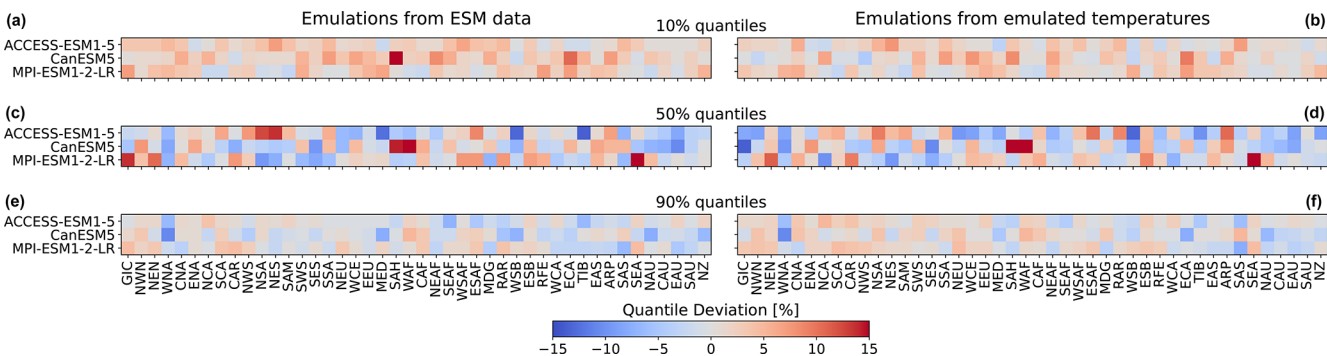

**Figure B5.** Same as Fig. 5 but for July.

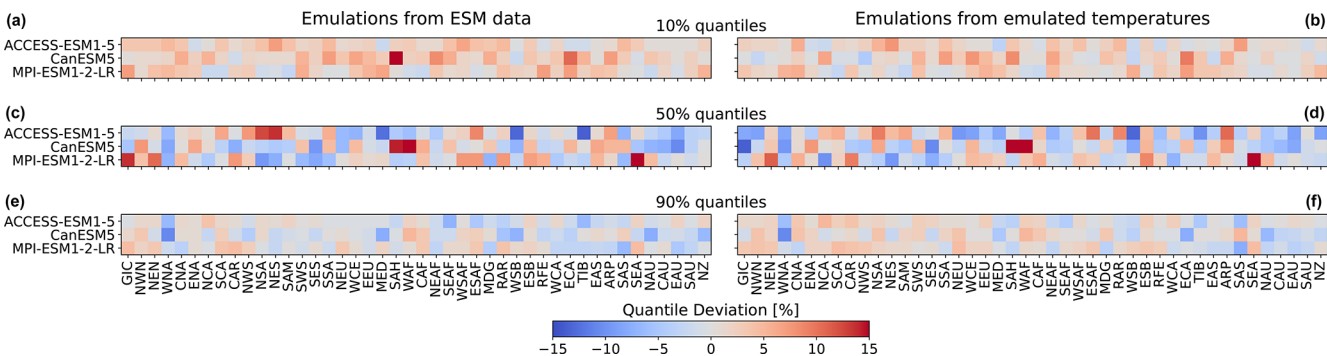

**Figure B6.** Same as Fig. 6 but for July.

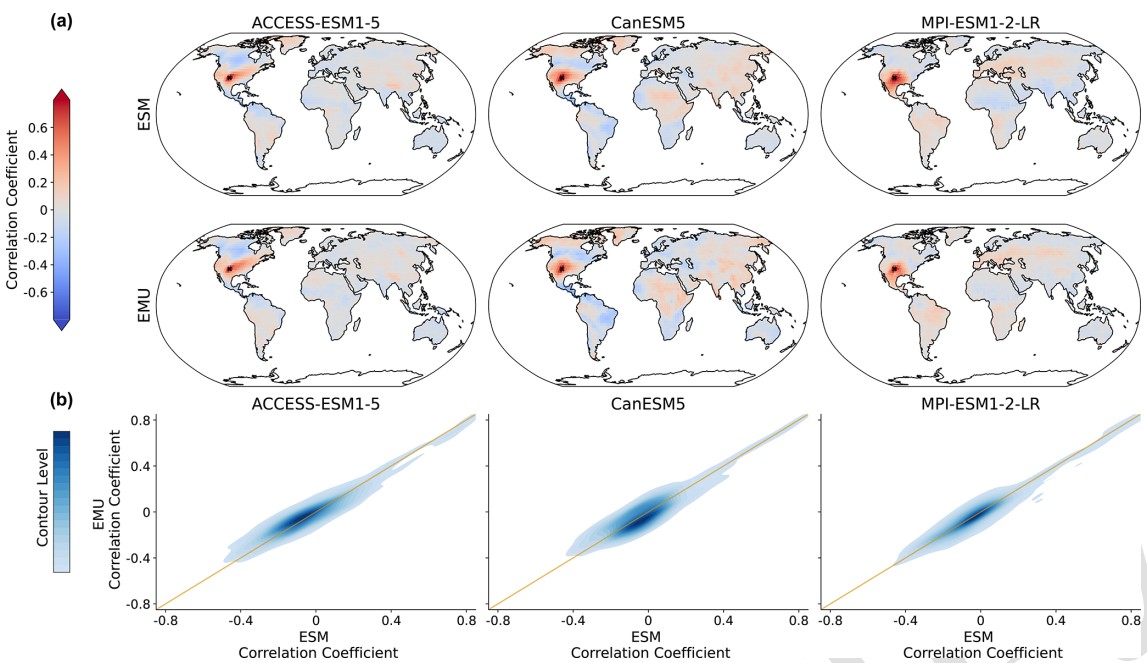

**Figure B7.** Same as Fig. 8 but for July.

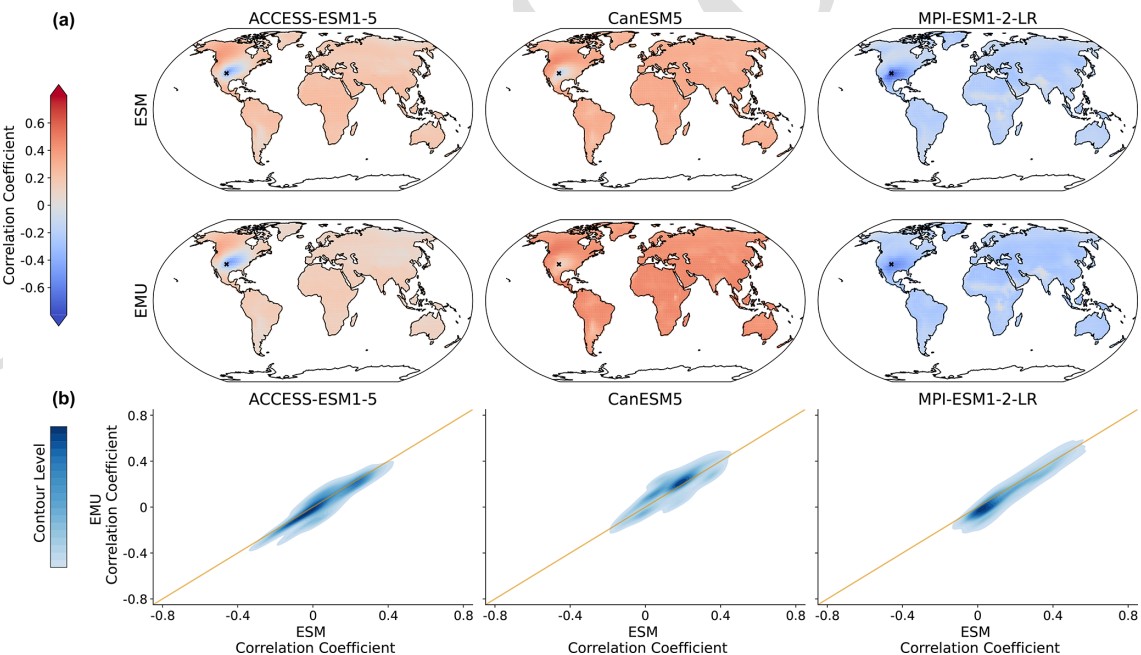

**Figure B8.** Same as Fig. 9 but for July.

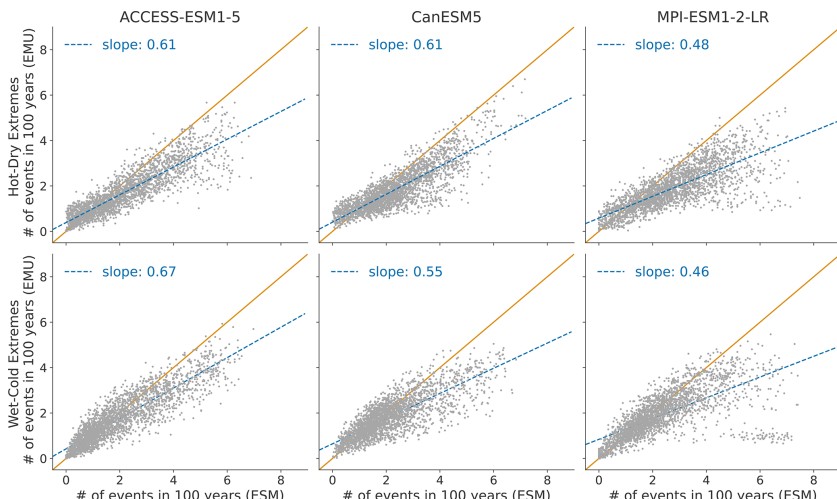

**Figure B9.** Same as Fig. 10 but for July.

## B3   Additional spatiotemporal validation metrics

In this section, we further verify the spatiotemporal char-
acteristics of the precipitation field at grid point level. In
Fig. B10, we assess the spatiotemporal characteristics at lo-
calised areas around chosen grid points using variograms.
A variogram measures the semivariance between grid points
as a function of distance. A semivariance of zero indicates
no difference between two points, with increasing values
for increasing dissimilarity between the time series at two
grid points. The emulator performs very well across mod-
els and regions for distances smaller than 500 km. At dis-
tances greater than 500 km, the emulator usually overesti-
mates semivariance (underestimates correlations) between
different grid points, with the differences between the ESM
data and emulations being most pronounced around the lo-
cation in NEU. These deviations are well within the inter-
model differences.

  We verify temporal characteristics at the same four loca-
tions using periodograms. A periodogram is an estimate of
the spectral density of the signal; that is, it gives an estimate
of how much power a signal has at each frequency. Gener-
ally, the emulator does well in approximating the grid point
level temporal characteristics. In all cases, the periodicity of
the annual cycle is pronounced and similar, although there is
mostly smaller pronunciation occurring at half-yearly or sea-
sonal intervals. In some cases, the emulator tends to overes-
timate white noise (NEUACCESS-ESM1-5 and CanESM5;
SEA ACCESS-ESM1-5). This suggests that we introduce
too much additional variability in the signal, and this could
have multiple reasons. For example, it might be that we oc-
casionally underestimate the trend, or it could also be that the
combined variability inherited from temperature through the
GLM and from sampling the multivariate stochastic process
superposes in some cases and leads to too much noise.

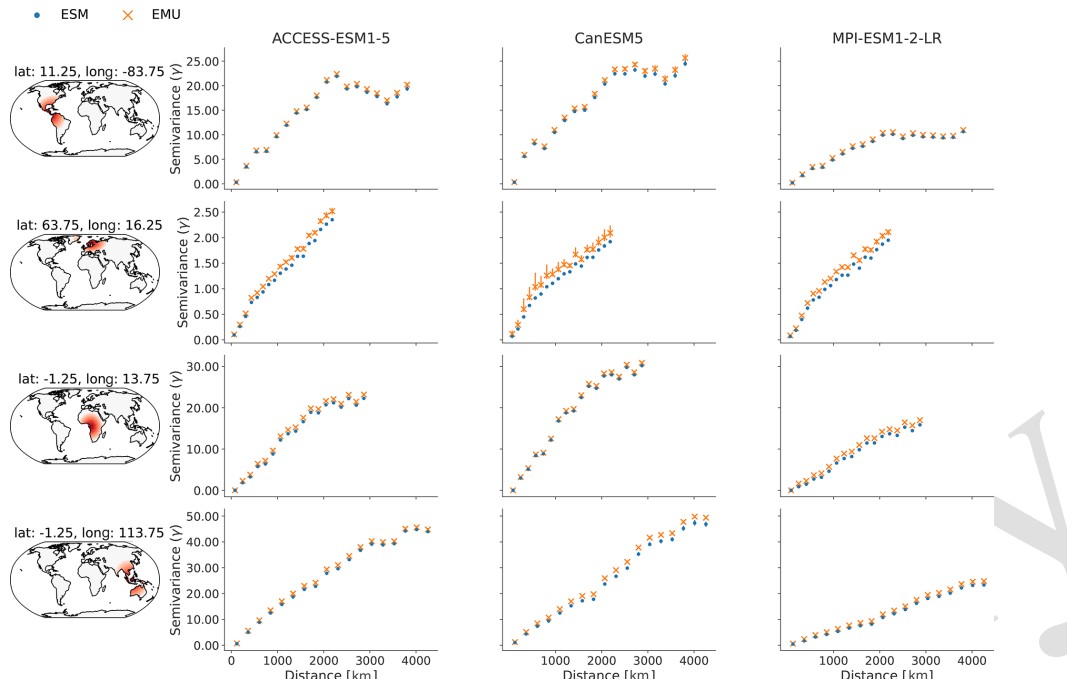

**Figure B10.** Variograms at four randomly selected grid points in SCA (lat 11.25, long −83.75), NEU (lat 63.75, long 16.25), CAF (lat −1.25, long 13.75), and SEA (lat −1.25, long 113.75) and the 300 closest grid points around the location. The first column indicates the selected grid points and neighbouring locations in shades of red and orange. The second, third, and fourth columns (ACCESS-ESM1-5, CanESM5, and MPI-ESM1-2-LR) show the median semivariance values and 10th–90th uncertainty ranges as vertical lines estimated from ESM data (blue) and EMU data (orange) at the selected grid points. Large values indicate dissimilarity between two grid points at the given distance, while small values indicate similarity. Note that the semivariance in the third and fourth rows (CAF and SEA) is large, such that 10th–90th quantile estimates are often contained within the marker size and are therefore not visible.

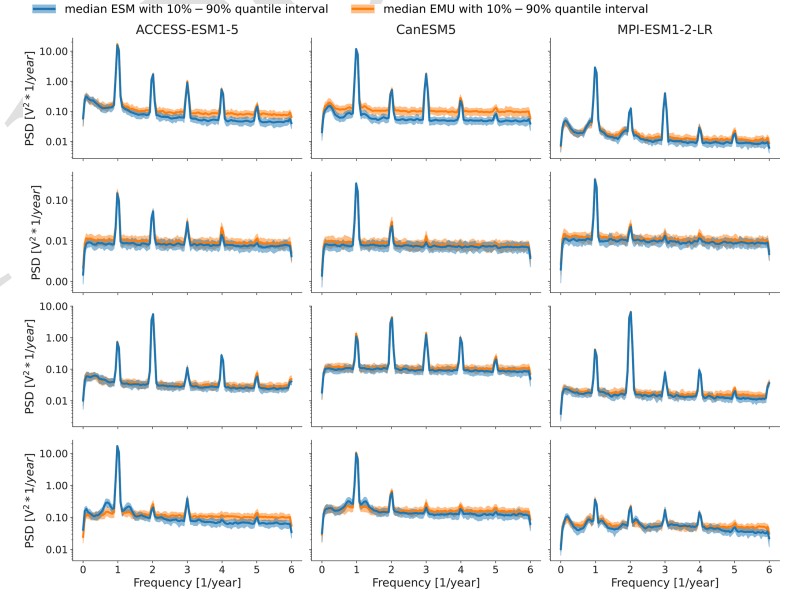

**Figure B11.** Periodograms for the same grid points as in Fig. B10 (top to bottom), with SCA (lat 11.25, long −83.75), NEU (lat 63.75, long 16.25), CAF (lat −1.25, long 13.75), and SEA (lat −1.25, long 113.75). Results show the median periodograms, along with 10th–90th uncertainty ranges, estimated from ESM data (blue) and EMU data (orange) at the selected grid points. At frequencies where only the ESM (blue) line is visible, the EMU and ESM estimates overlap.

## Appendix C: Forcing MESMER-M-TP with emulated temperatures

MESMER-M-TP is conceived as a module that can be used in conjunction with existing temperature emulators (e.g. Nath et al., 2022). The idea of this section is to understand and assess the performance of the emulated precipitation fields in such a coupled setting and to verify that our emulation approach is robust with respect to the gridded temperature input. The coupled emulation framework validated in this section translates the global mean temperatures into gridded temperatures and the gridded temperatures subsequently into precipitation field time series with coherent cross-variable characteristics, meaning that the coupled emulator allows us to go from global mean temperatures to joint temperature and precipitation data.

### C1    Generating a dataset of emulated temperatures

We generate temperature emulations using global mean temperature as a driver as follows. First, the temperature field is projected onto its principal components (PCs). Next, we decompose the global mean temperature (GMT) into a trend and a variability component, as suggested in Beusch et al. (2020). We then fit a linear model to each principal component individually with the trend and the variability in global mean temperature (GMT) as the two sole forcing variables. We then compute the residuals as the difference between the original principal components and the linear fit. Next, we use a Yeo–Johnson transform to ensure the residuals follow a normal distribution. Subsequently, we approximate the residuals as an auto-regressive (AR) process of the of order 1, with the AR coefficients varying by month, as suggested in Nath et al. (2022). We calibrate the model parameters following our calibration approach for precipitation. We use a single ESM ensemble member across SSPs for training and use all other ensemble members for testing (see Sect. 3.1). Following Beusch et al. (2020), we generate additional realisations of GMT. We then drive the linear model with the new GMT realisations to get trend estimates of the PCs and draw new samples from the AR(1) process to emulate variability. Last, we add the trend estimates to the variability samples and apply the inverse of the PCA to get a set of emulated temperatures. For the validation section, we generated 100 temperature emulations for each model. In Figs. C1 and C2, we exemplarily show the time series of emulated temperature data and actual ESM temperature. An indication for the quality of the emulations is the quantile deviations shown in Figs. C3 and C4. The emulation approach works well, although temperatures are slightly underdispersive, similar to results in Beusch et al. (2020); Nath et al. (2022). In any case, we are mainly interested in the joint emulation error and the robustness of emulated precipitation results with respect to emulated temperature input.

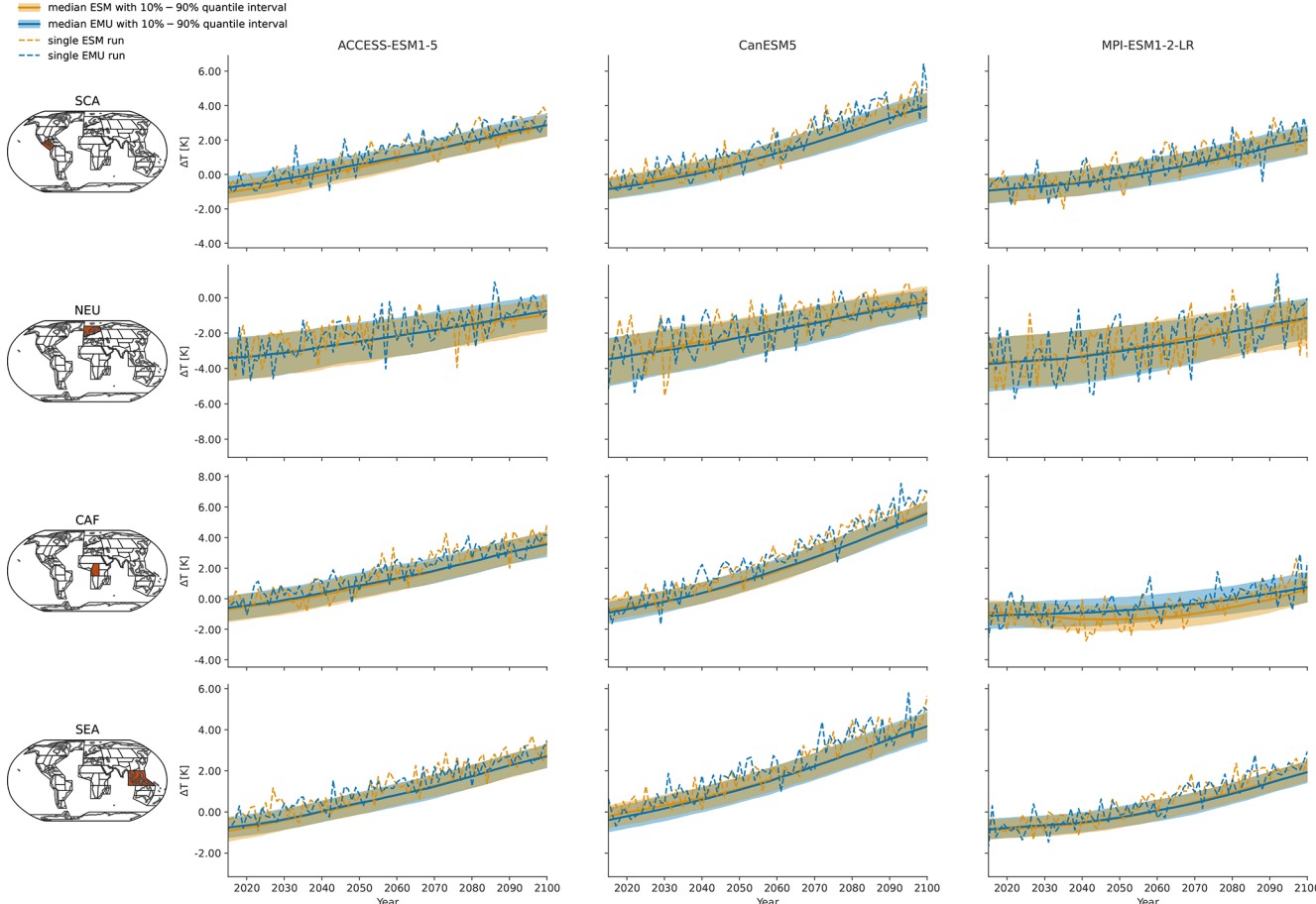

**Figure C1.** Same as Fig. 3 but for temperatures in January.

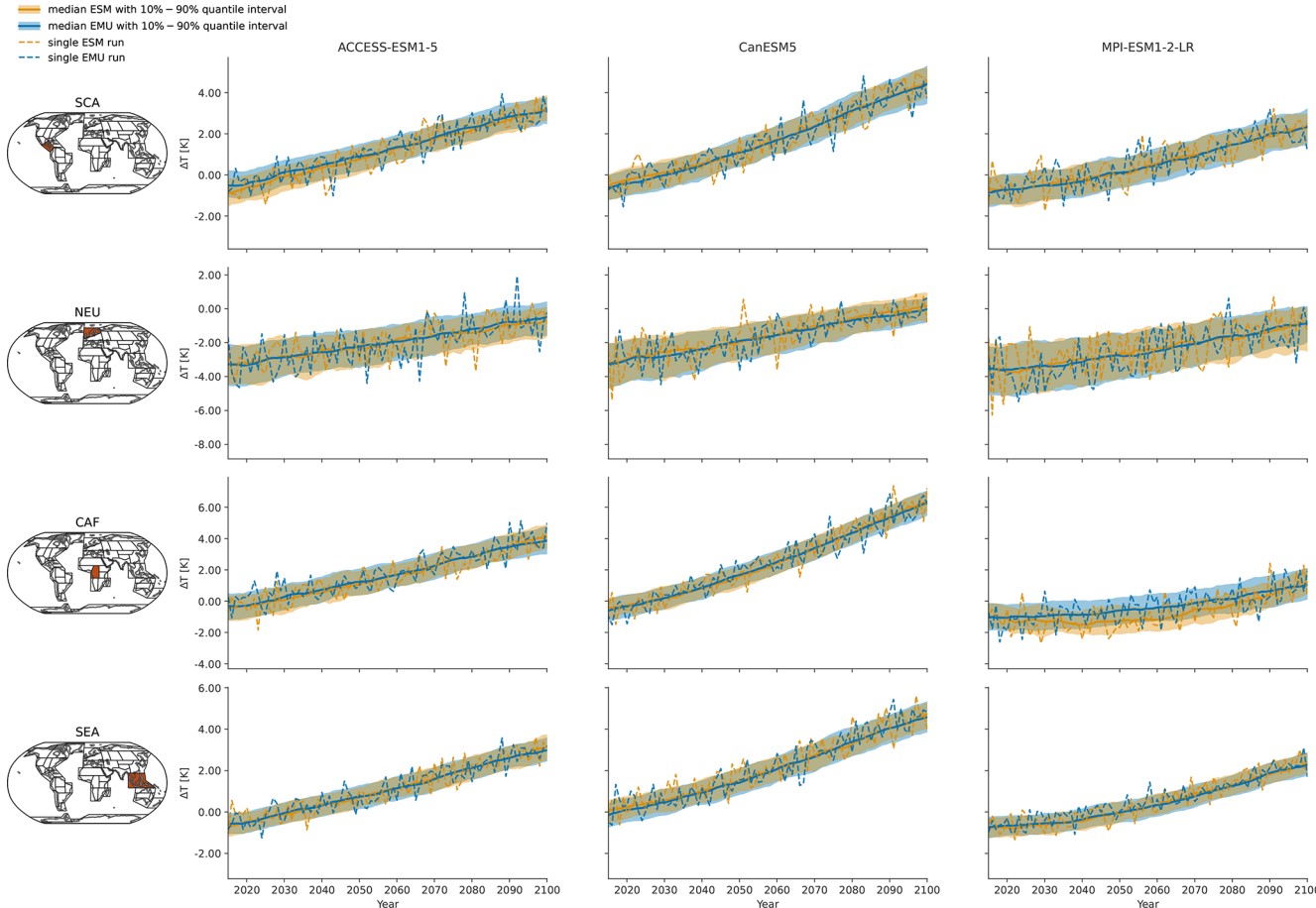

**Figure C2.** Same as Fig. 3 but for temperatures in July.

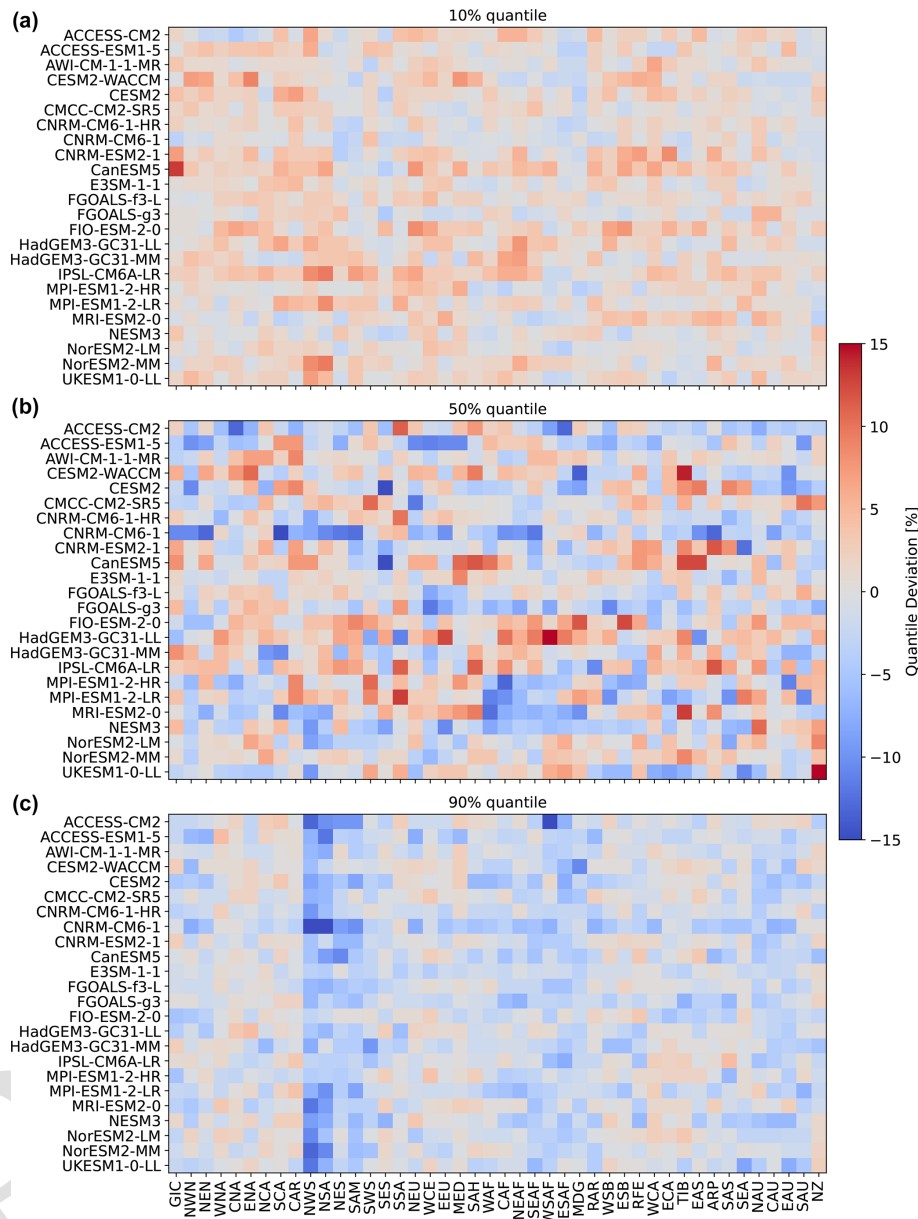

**Figure C3.** Same as Fig. 6 but for temperatures in January across all 24 models.

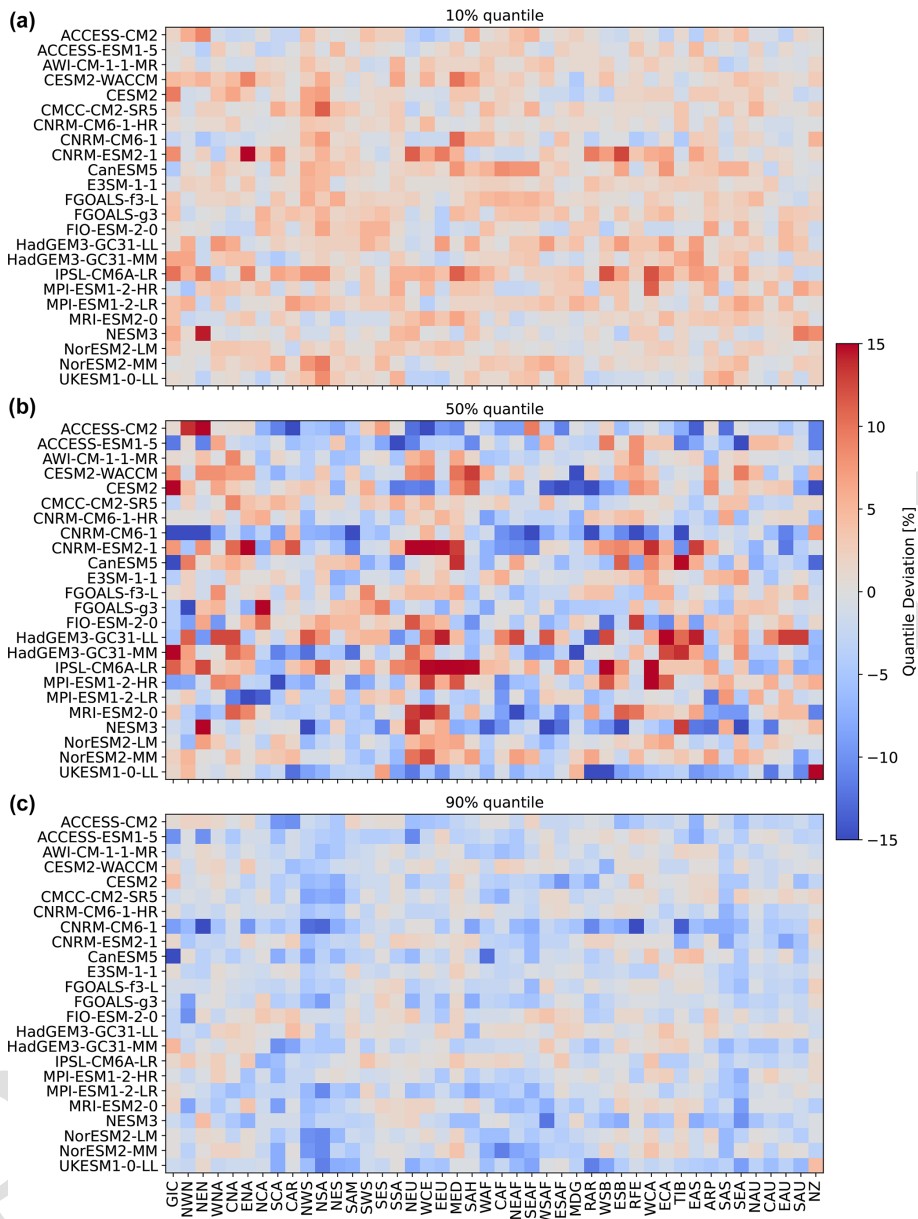

**Figure C4.** Same as Fig. 6 but for temperatures in July across all 24 models.

## C2   Results for coupled emulations

This section presents all results shown in the main paper, as well as in Sect. B2, for the coupled results.

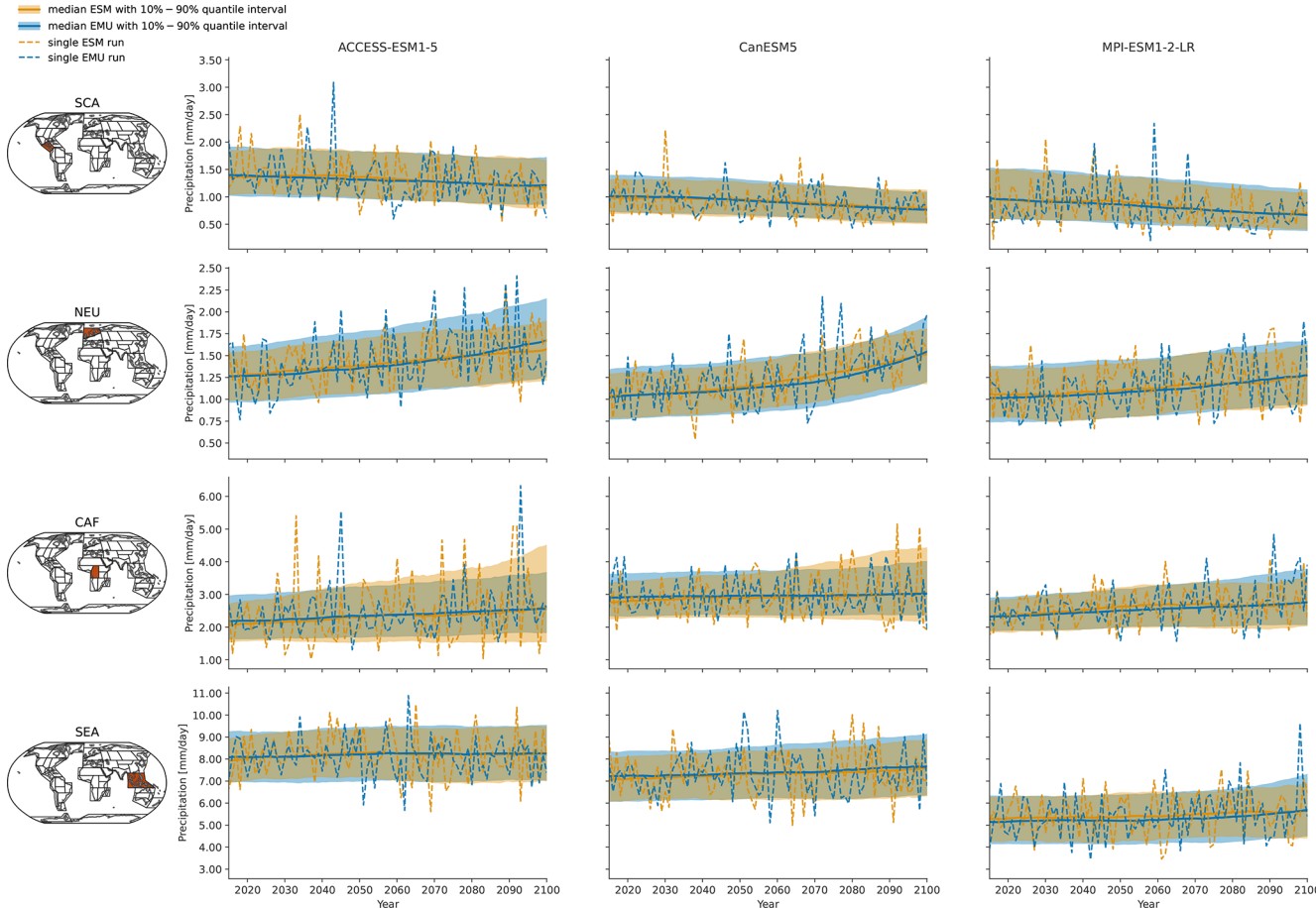

**Figure C5.** Same as Fig. 3 but for the coupled emulations in January.

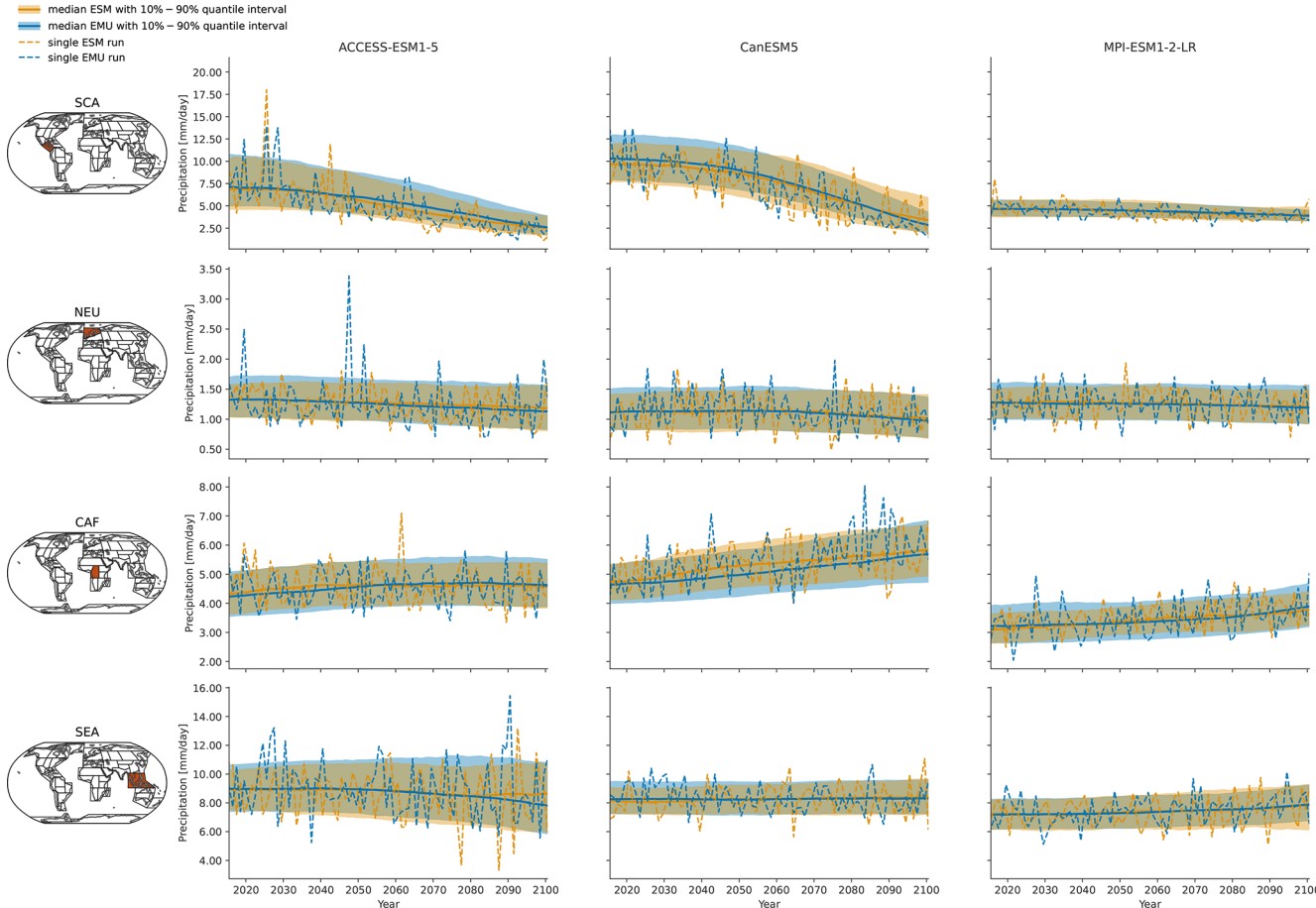

**Figure C6.** Same as Fig. 3 but for the coupled emulations in July.

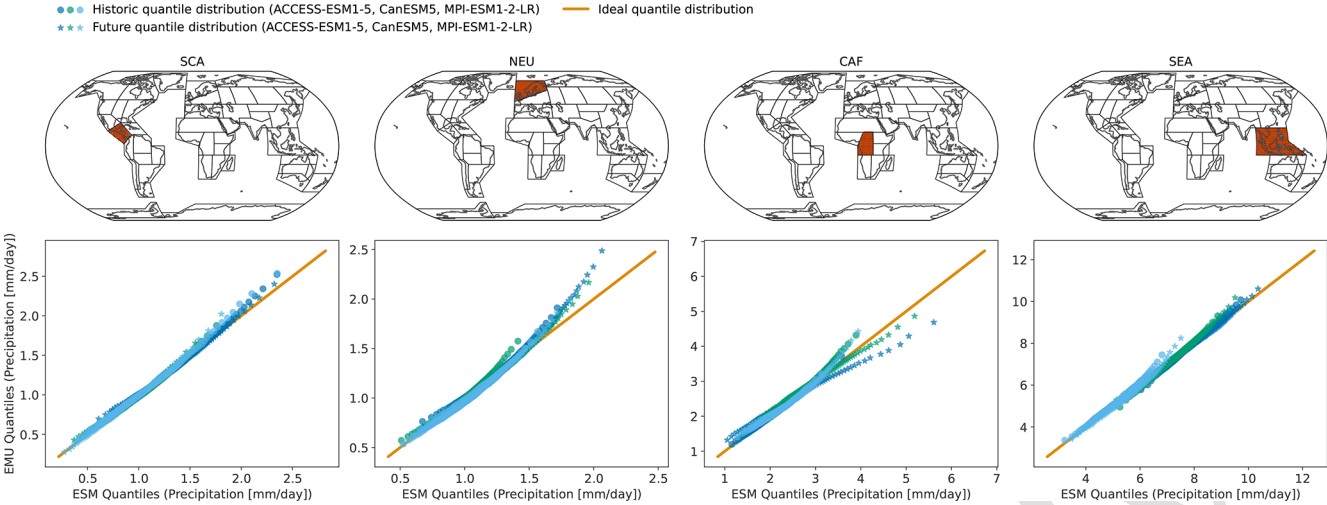

**Figure C7.** Same as Fig. 4 but for the coupled emulations in January.

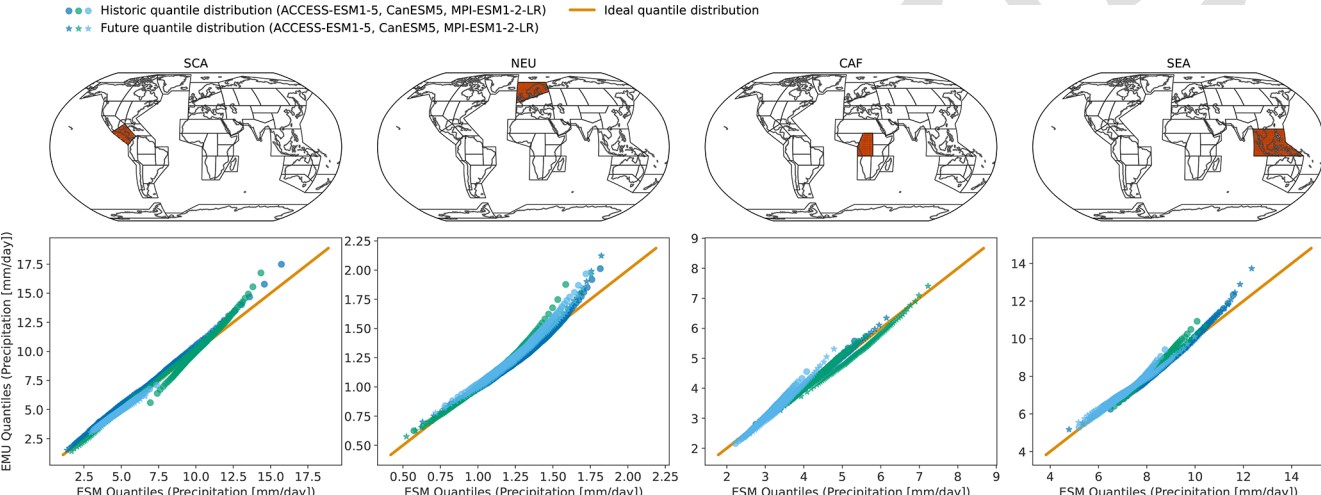

**Figure C8.** Same as Fig. 4 but for the coupled emulations in July.

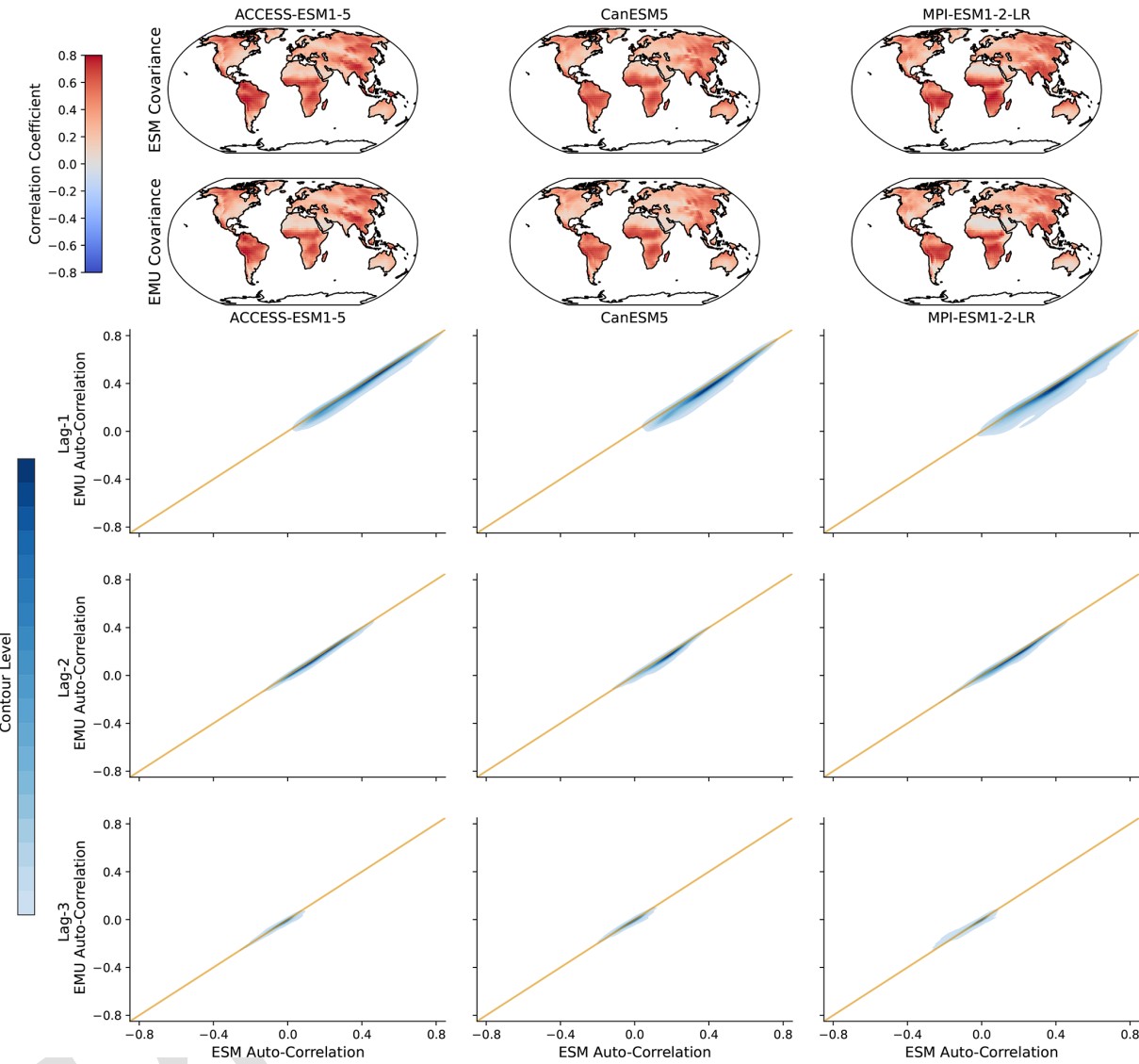

**Figure C9.** Same as Fig. 7 but for the coupled emulations.

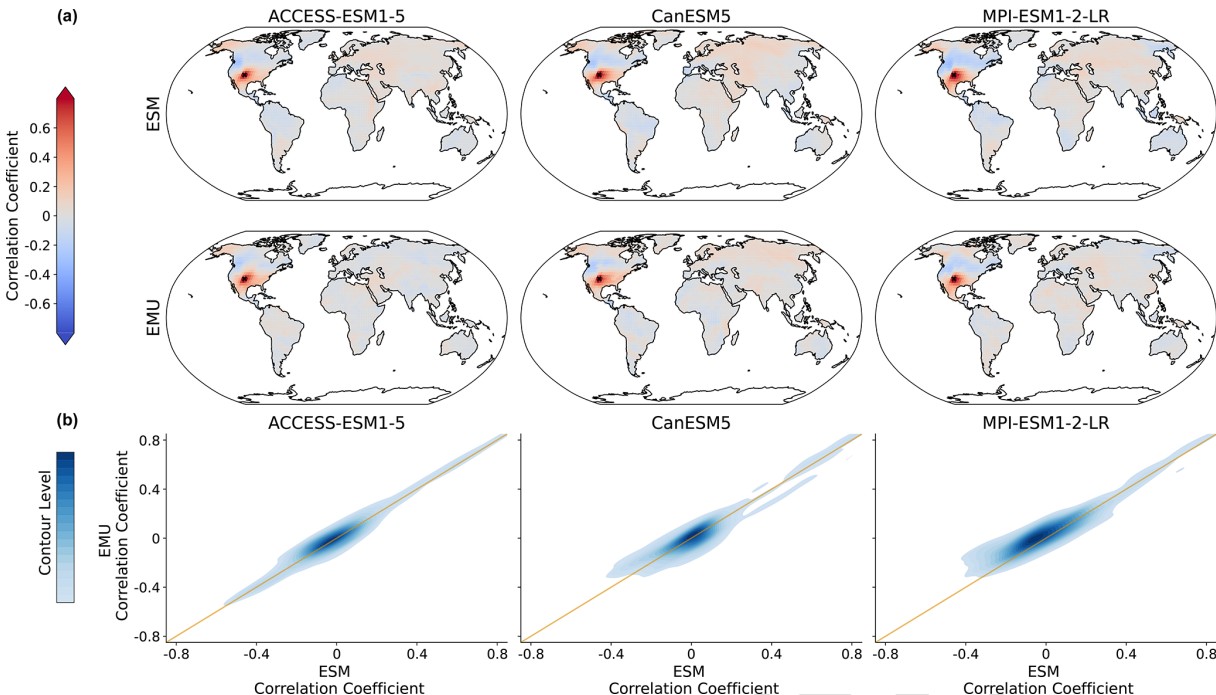

**Figure C10.** Same as Fig. 8 but for the coupled emulations in January.

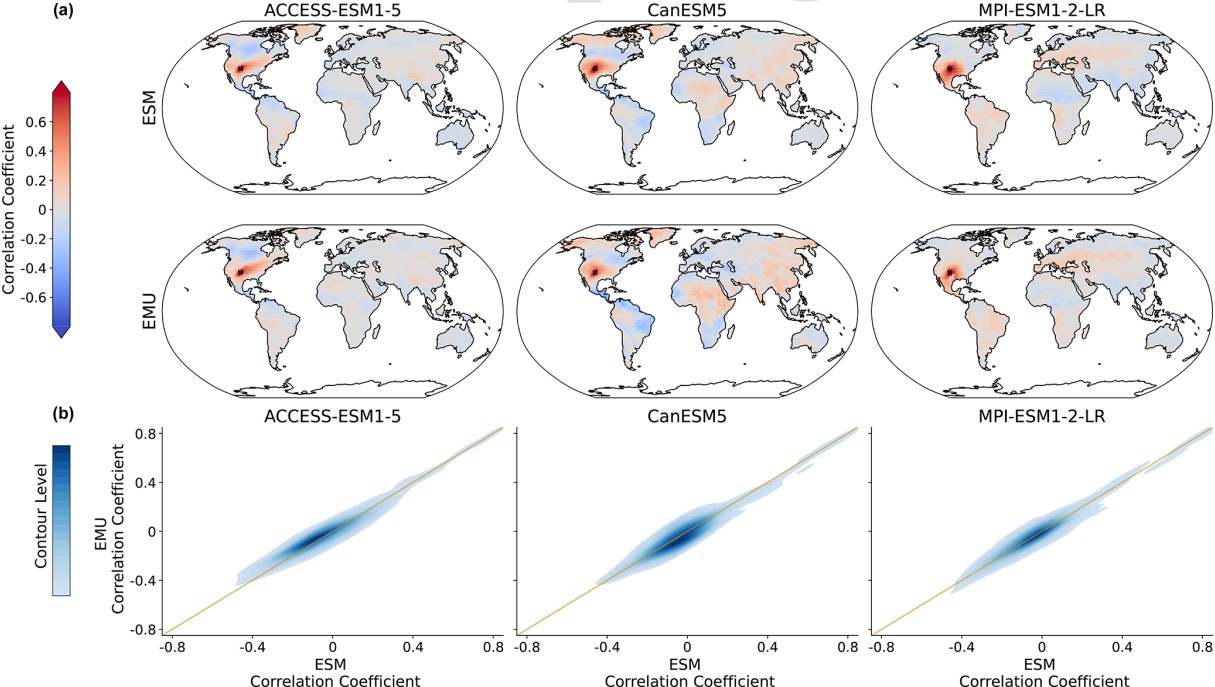

**Figure C11.** Same as Fig. 8 but for the coupled emulations in July.

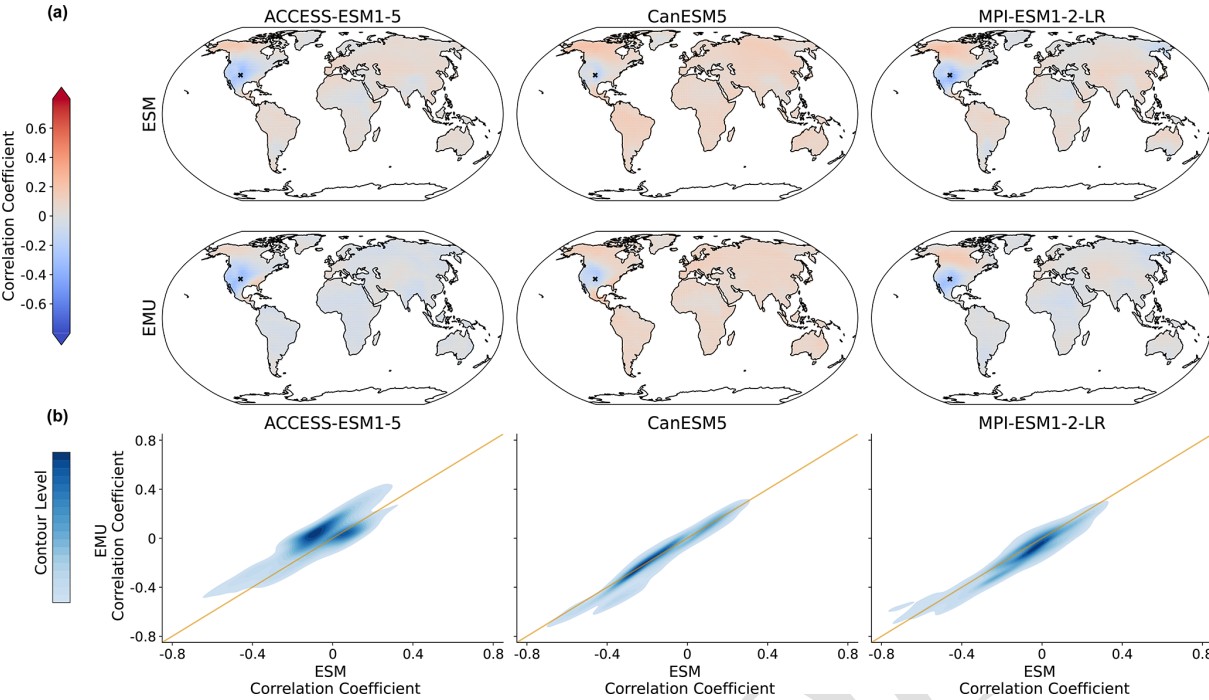

**Figure C12.** Same as Fig. 9 but for the coupled emulations in January.

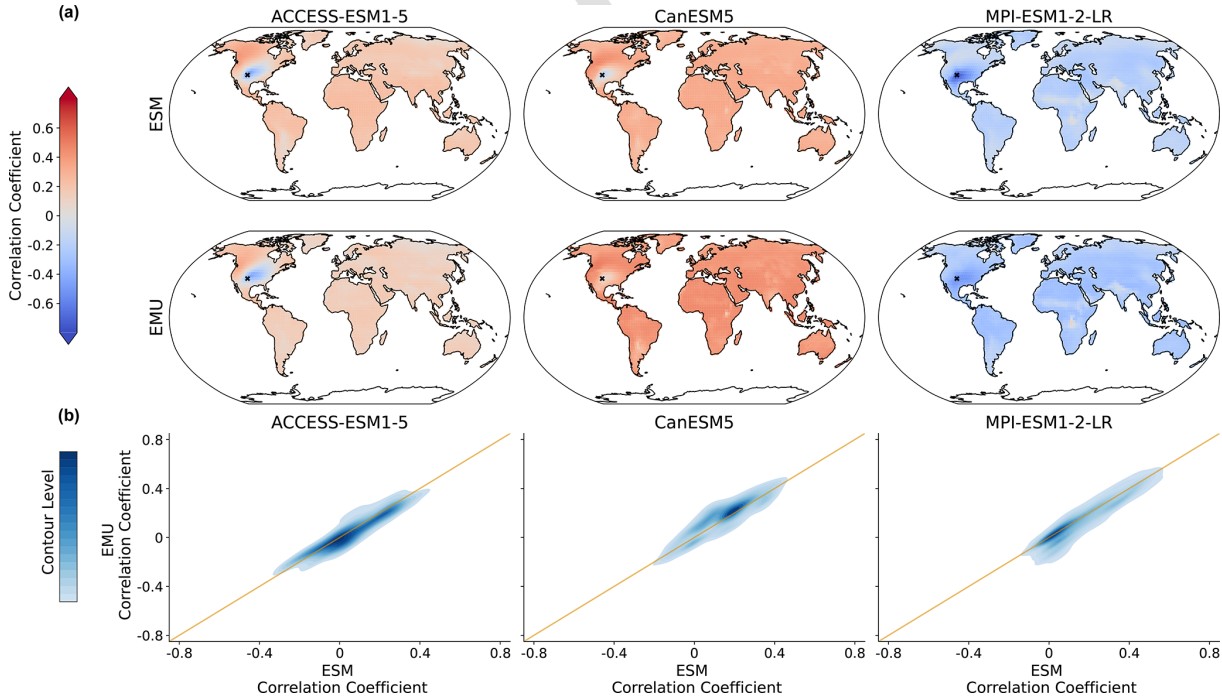

**Figure C13.** Same as Fig. 9 but for the coupled emulations in July.

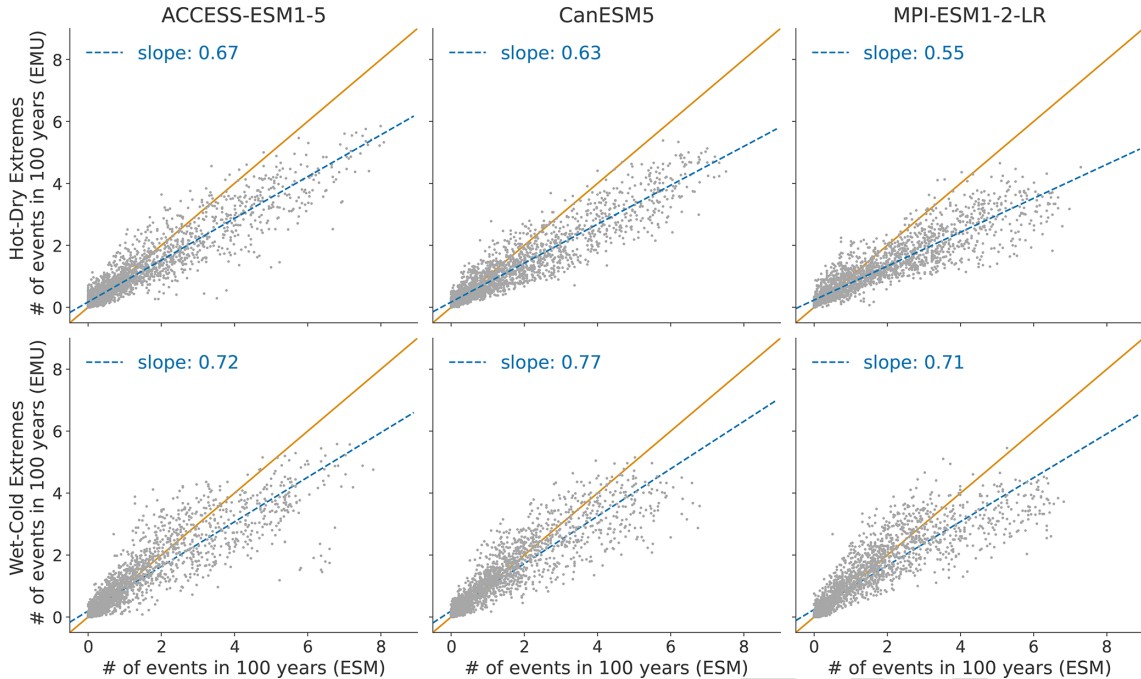

**Figure C14.** Same as Fig. 10 but for the coupled emulations in January.

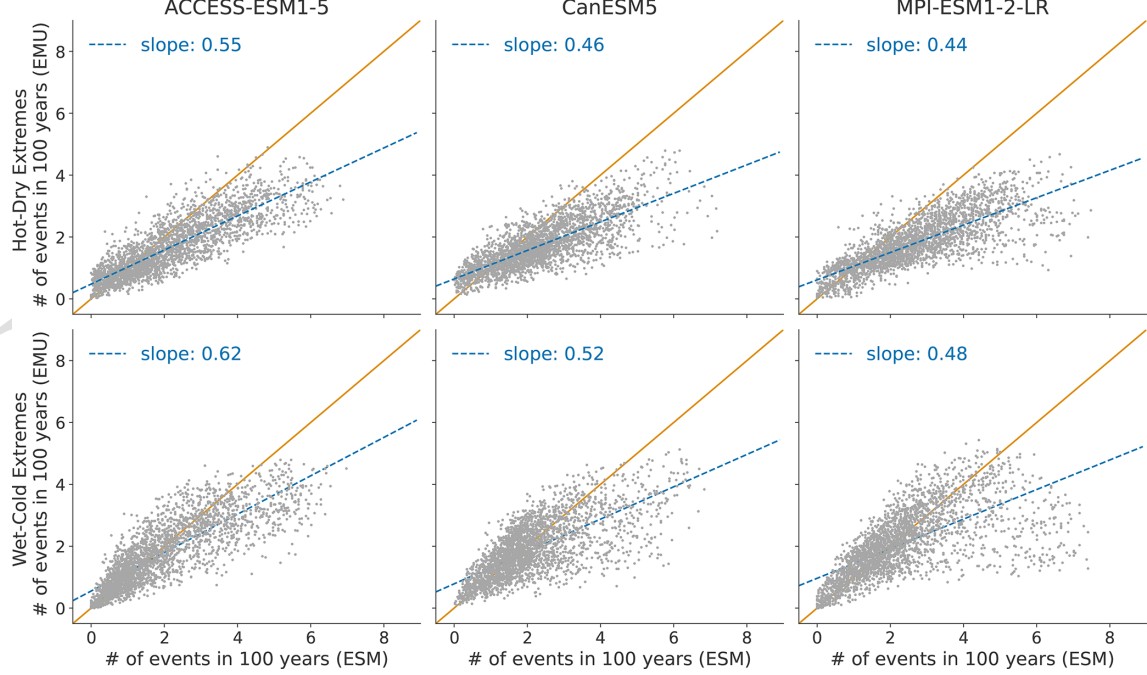

**Figure C15.** Same as Fig. 10 but for the coupled emulations in July.

*Code and data availability.* The current version of the model MESMER-M-TP is available on GitHub at https://github.com/sarasita/mesmer-m-tp (last access: 12 November 2024). The exact version of the model used to produce the results used in this paper is archived on Zenodo (https://doi.org/10.5281/zenodo.11086167, sarasita, 2024). In addition, code for MESMER and MESMER-M can be found at https://github.com/MESMER-group/mesmer (last access: 12 November 2024). Whenever MESMER(-M) data were used in this study, we relied on MESMER v0.9.0, which is available on Zenodo (https://doi.org/10.5281/zenodo.10408206, Hauser et al., 2023). The analysis can be reproduced using the code on Zenodo and using ESM data, as described in Brunner et al. (2020) and available from the public CMIP archive at https://esgf-node.llnl.gov/projects/cmip6/ TS4 .

*Author contributions.* SaS, LG, SIS, and CFS conceived the study. SaS developed the methods, with contributions from LG and PP. SaS wrote the paper, with contributions from all authors.

*Competing interests.* The contact author has declared that none of the authors has any competing interests.

*Acknowledgements.* The authors want to thank the two anonymous reviewers for their valuable comments which helped make this study more comprehensive.

*Financial support.* This research has been supported by the Deutsche Bundesstiftung Umwelt (Promotionsstipendium) and the EU HORIZON EUROPE Framework Programme, EU HORIZON EUROPE Innovative Europe (grant nos. 101003687 and 101081369). TS5

*Review statement.* This paper was edited by Po-Lun Ma and reviewed by Chih-Chi Hu and one anonymous referee.

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
