# Peer review of "Introducing the MESMER-M-TPv0.1.0 module: Spatially Explicit Earth System Model Emulation for Monthly Precipitation and Temperature"

_EGUsphere, 2024_

## Referee Comment (RC2)

**Summary & General comments:**

The MESMER-M-TP v0.1.0 module is designed to emulate spatially explicit monthly mean precipitation fields using monthly mean temperature fields as predictors. It offers runtime efficiency, which facilitates the efficient exploration of the precipitation forecast and its uncertainty under different scenario or parameters. Despite for some misses in extreme events, the emulator can effectively generate spatially coherent statistics of temperature and precipitation time series in general.

Overall, I enjoyed reading this paper very much. I appreciate that the emulator is carefully designed with physical knowledge and intuition. I also value the simplicity of the emulator, making it easy to understand and interpret. In terms of writing, the methodology is described in detail, the results are well-presented with sufficient explanation of the calculations. The significance and the implications are also discussed. Although I think the paper is already in a good shape, I do have a few minor comments below that I hope can further improve its quality and readiness.

**Specific comments:**

L6: MESMER(-M): I suggest put the full name here

L94: row-vector: isn't it a column vector?

L103: the local precipitation... -> I suggest "the emulated local precipitation"

A general comments for notations and equations: Please make sure all the notations are used and defined in a consistent way.

e.g.,

(1) Equations (3)-(6):

I assume

$$E[P_{s,m}|\mathbf{X}_{s,m}] = E[P_{s,m}|X_{s,m}^T] = \mu_{s,m}$$

$$V[P_{s,m}|\mathbf{X}_{s,m}] = var_{s,m}$$

$$k_{s,m} = \Phi_{s,m}$$

I suggest avoiding the use of multiple notations for the same variables, which might cause confusion.

Also, is the dispersion $k_{s,m} = \Phi_{s,m}$ a constant for different {s,m}? Please clarify.

(2) Equation (8):

Is $f_{s,m}(X_{s,m}^T) = E[P_{s,m}|\mathbf{X}_{s,m}]$? Please clarify as the form of $f_{s,m}$ is not defined elsewhere in other equations. Also, $X_{s,m}^T$->$\mathbf{X}_{s,m}$ to be consistent.

(3) Figures 7&8: $Pr_{s,m}$ -> $P_{s,m}$ to be consistent with the Equations

(4) L154: the first principal component $PCA_{s,m}^1$ -> should be $PCA_{s,m}^0$?

(5) Equation (7): last column $PCA_{m,s,y_0}^{p-1}$ -> $PCA_{s,m,y_0}^{p-1}$

L170-175: I suggest describing the methodology in more detail, perhaps including a few equations for clarity.

L213: How sensitive are the results to the number of closest grid points n ? Does using PCA to define the predictor make the results less sensitive to the choice of n?

L218-219: I suggest including an equation to show the actual form of the prediction model, including the interaction terms used in this study.

L243-249: My understanding is that, given the temperature field around a grid point, the emulator predicts a probability distribution of precipitation at the point rather than a single predicted value. Is that correct? If so,

(1) Do you actually draw random samples from the pdf to characterize the distribution? If so, how many samples are drawn?

(2) How do you verify the EMU results (which is a distribution) against ESM (which is a single value)?

(3) What exactly does the quantile refer to in Figure 2? I assume for the ESM, the quantiles are calculated from the ensemble members (so sample size = # of ensemble member), but what about the EMU results (what are the samples size for the quantiles)?

L296: Fig.3 and B1 -> should be B2?

A general suggestion for the figures: it might be helpful to add subtitle like (a),(b),(c) for subplots in some figures. This can make the figure captions easier to describe and the references to the figure in the text clearer.

Figure 2 - L4: Orange/Blue lines represent precipitation estimates of a single ESM/EMU ensemble member -> should be "dashed" lines to distinguish with the solid lines

L300-302: This is an assumption in the modeling framework, is that correct? If so, I suggest change "suggests" -> "assumes"

L314-316: It is not immediately clear to me why this suggests that the model struggles to disentangle the trend and variability?

---

## Author Comment (AC1)

**Authors Response to Referee #1**

We thank referee #1 for all the time they invested in reading and understanding this paper and in particular, for their constructive comments and suggestions to the manuscript we have submitted. We took all the comments into consideration for generating a revised version of the manuscript. We respond to the original comments (italic) point-by-point.

*This emulator, a new addition to the MESMER family, produces fields of monthly precipitation as a function of corresponding fields of monthly average surface temperature. It does so by modeling precipitation at each grid point as the product of a mean component, driven by temperature in a neighborhood of the location, and a residual component whose variability is estimated as constant along the length of the simulation.*

*The results, for both the case when temperature is the output of the ESM being emulated, and the case when it is in turn a product of emulation by the mean component of MESMER-M, are validated according to numerous metrics that address first and higher order behavior, trends, spatial and temporal coherence within the precipitation output and between precipitation and temperature, with satisfactory results.*

*I enjoyed reading this paper, which is clearly written and presents a wide range of validation results creatively conceived and displayed. The authors have a clear-eyed take on the outcomes, which are not perfect but constitute an important step forward in emulating impact relevant variables. This is especially true for an emulator that can generate novel realizations, akin to additional initial condition ensemble members, and therefore can help characterize statistics of extremes (within the limits of what ESMs and their emulator can represent, through monthly average quantities). As the authors recognize, there is room for improvement, especially for the highest quantiles of precipitation which often appear overestimated when the validation is done within the confines of an individual model statistics, but these errors pale compared to the inter-model variability/differences. It is also good that the authors are able to offer an hypothesis about the sources of the emulator shortcomings (having to do with the variability term being modeled as stationary rather than allowing it to vary with temperature changes as well) and therefore can point at directions for further development.*

AR: Thank you for this very accurate summary of our work.

*My comments are in the spirit of creating a slightly more comprehensive picture of the emulator-ESM comparisons (again recognizing the nice effort in providing a multidimensional depiction of the emulator performance, always hard to do). What I would have liked to see are:*

- *Maps of gridded trends and patterns of change for a few time slices along the century, comparing emulated and ESM projections, maybe as seasonal averages. These could give an additional sense of the mean/trend performance at the grid-point level, performance evaluation which is currently - for these aspects -- limited to aggregated regional (SREX) scales.*

AR: We think showing seasonal estimates of gridded output would be very useful. We have included Figure 2 (and Figures B1 and B2) in the revised manuscript.

- *Additional analyses of spatial and temporal characteristics of the generated precipitation fields. These could be performed for a set of locations, chosen to represent different latitudes/climate conditions. These analyses could include variograms, which are the tool of choice for analyzing the spatial correlation characteristics of a field from spatial statistics, and could be computed for the regions around each location. Similarly, for time series of output at specific locations one could look at the entire autocorrelation structure, or spectrum, and have a full picture of what the emulator does well or misses (for example, ENSO-driven oscillations in precipitation at points known to have strong teleconnection, and of course compared to what the ESM does, since it is not a given that ESMs represent those signals well. Hopefully at least one of these three ESMs has a decent ENSO behavior and it would be interesting to see if the emulator can represent it).*

AR: We agree with all the suggestions. As we want to avoid displaying too many verification metrics in the main of the paper, we have addressed the suggestions in Appendix B3 called "Additional Spatio-temporal Validation Metrics" (L434 ff.). The main analysis steps we took are the following:

1. Geo-spatial statistics: Figure B10 now displays variograms that focus on 4 locations and their 300 closest grid-points. The locations were selected at random within the regions SCA, NEU, CAF, SEA. These regions are consistent with the validation approach employed throughout the entire paper and represent a wide range of possible precipitation behaviors. We discuss the results in L440-446.
2. Temporal characteristics: For the same four locations, we generated the full frequency-periodograms and included the results in Figure B11. Periodograms decompose the power of the precipitation signal into individual frequency components. The findings are described in L446-454.

3. Lastly, we also tried to incorporate your suggestions on ENSO behavior. The main difficulty being that most established ENSO indices we are aware of rely on ocean data while we emulate land data only. We nevertheless used MPI-ESM1-2-LR, a model that depicts ENSO feedback reasonably well [1], to perform experiments identifying ENSO signals over land. To this end, we jointly decomposed the gridded temperature and precipitation fields into their principal components independently for ESM and EMU data. Below, we are showing the correlation coefficients between the first four principal components (top to bottom) and January ESM/EMU temperature (1st/2nd column) and ESM/EMU precipitation (3rd/4th column). We identify the 4th component to resemble ENSO behavior as expected in December, January and February (e.g. warming over North-Western North America, Eastern and South Asia, Eastern South Africa, North-West South America, Madagascar, Australia; cooling over Eastern North America; Dry conditions over Eastern South Africa, Madagascar; Wet conditions in Western, Central and Eastern North America, North Central America). In addition, we performed a frequency-analysis on the first four principal components. The results are displayed in the fifth column (ESM: blue lines, EMU: orange lines). The El Nino frequency bands (2-7 years) are highlighted for the third component using black vertical lines. The signal strength peaks at an occurrence of 0.33 (once every 3 years). The signal is present in the emulated data as well, although it is less pronounced. We appreciate that this analysis, despite showing some encouraging results, is limited as we do not have ocean data available as part of our emulation. Because of the limitations, we are hesitant to include the analysis on ENSO behavior in the main manuscript, but now mention it as an area for future development (L415-417).

   a. [1] Tian, Q., Ding, R., & Li, J. (2023). Simulations of the North Tropical Atlantic Mode–ENSO connection in CMIP5 and CMIP6 models. *Journal of Geophysical Research: Atmospheres*, *128*(16), e2023JD039018.

[Figure]

*My other overarching feedback is that some of the figures are very small and it is hard to distinguish the different points/color hues. Either expanding the area of the page dedicated to them, or reducing the range of some of the axis to focus on the region of the plot where something is happening could help (even if I realize that the same ranges may be needed for ease of comparison). But all this said, I appreciate the graphics that the authors have created to display these results.*

AR: We agreed and adjusted all graphics for better visibility. In particular, we adjusted all tick and label sizes and hope that the graphics are more easily accessible now.

*I am aware of another emulator proposed recently, for daily precipitation, that may be relevant as a citation (if only for its use of a Gamma distribution): Kemsley, S. W., Osborn, T. J., Dorling, S. R., & Wallace, C. (2024). Pattern scaling the parameters of a Markov-chain gamma-distribution daily precipitation generator, International journal of climatology, 44(1), 144–159. https://doi.org/10.1002/joc.8320*

AR: Thank you for catching this one. We have included it in our paper now.

*These are my only substantial comments, and I would be pleased to see a slightly revised version addressing them, but I consider this work in very good shape already so I would characterize my requests as minor revisions and I hope the authors can meet them with ease. Nice work.*

AR: Thank you again for taking the time to look at our paper so thoroughly and for the, overall, very supportive and positive feedback. We hope we could address your suggestions sufficiently in the revised manuscript.

---

## Author Comment (AC2)

**Authors Response to Referee #2**

We thank referee #2 for all the time they invested in reading and understanding this paper and in particular, for their constructive comments and suggestions to the manuscript we have submitted. We took all the comments into consideration for generating a revised version of the manuscript. We respond to the original comments (italic) point-by-point.

*Summary & General comments:*

*The MESMER-M-TP v0.1.0 module is designed to emulate spatially explicit monthly mean precipitation fields using monthly mean temperature fields as predictors. It offers runtime efficiency, which facilitates the efficient exploration of the precipitation forecast and its uncertainty under different scenario or parameters. Despite for some misses in extreme events, the emulator can effectively generate spatially coherent statistics of temperature and precipitation time series in general.*

*Overall, I enjoyed reading this paper very much. I appreciate that the emulator is carefully designed with physical knowledge and intuition. I also value the simplicity of the emulator, making it easy to understand and interpret. In terms of writing, the methodology is described in detail, the results are well-presented with sufficient explanation of the calculations. The significance and the implications are also discussed. Although I think the paper is already in a good shape, I do have a few minor comments below that I hope can further improve its quality and readiness.*

AR: Thank you very much for this assessment. We found all your comments very helpful and appreciated the precise suggestions. We find that implementing them in the revised manuscript has overall increased the quality of our paper.

*Specific comments:*

*L6: MESMER(-M): I suggest put the full name here*
*L94: row-vector: isn't it a column vector?*
*L103: the local precipitation... -> I suggest "the emulated local precipitation"*

AR: All of them were very good catches and we have adjusted the manuscript accordingly.

*A general comments for notations and equations: Please make sure all the notations are used and defined in a consistent way.*

*E.g., (1) Equations (3)-(6); (2) Equation (8); (3) Figures 7&8; (4)L154, (5)Equation(7)*

AR: We are very sorry for the confusion regarding the notation. We did change the notation multiple times during the writing of this paper (for clarity). It seems that we accidentally mixed the different notations on multiple instances. We incorporated all your suggestions in our paper and carefully checked the rest of the manuscript for inconsistencies. We hope that the new set of equations is now concise. (L89ff)

*L170-175: I suggest describing the methodology in more detail, perhaps including a few equations for clarity.*

AR: We are describing the Kernel Density Estimation and the subsequent sampling approach in more detail now and to this end also included two equations (L165-182). We hope that the new description is clearer now.

L213: How sensitive are the results to the number of closest grid points n? Does using PCA to define the predictor make the results less sensitive to the choice of n?

AR: The results are rather robust with respect to the number of closest grid points. We have tested for n between 75 and 400. Larger n become too computationally expensive, while setting n=2652 results in a single PCA for the entire globe which strongly varies with region. The main reason for including the PCA is the strong spatial correlation of temperature, which leads to difficulties when fitting a Linear Model. We have tried to overcome these difficulties by using Lasso/Ridge/ElasticNet regression (rather than performing a Regression on the PCA transformed temperatures). Lasso Regression selects data from only a few grid-points to build a regression model, making the precipitation estimates overly sensitive to the temperature field, while Ridge Regression proved to be very sensitive to the training dataset during cross-validation. We found the solution relying on the PCA a good compromise. We have added some of this information to the manuscript (L219-227).

*L218-219: I suggest including an equation to show the actual form of the prediction model, including the interaction terms used in this study.*

AR: Great suggestion. We introduced equation (10) (~L242)

*L243-249: My understanding is that, given the temperature field around a grid point, the emulator predicts a probability distribution of precipitation at the point rather than a single predicted value. Is that correct? If so,*

AR: Your understanding is exactly right.

1. (1) Do you actually draw random samples from the pdf to characterize the distribution? If so, how many samples are drawn?

AR: We approximate the empirical distribution of the residuals using a Kernel Density Estimation (KDE) and then, we draw new samples from the fitted distribution to generate additional realizations of variability. We rely on a 1-on-1 coupling meaning we sample a single time series from the KDE for each gridded temperature field we use as forcing. This implies that when we force the emulator with gridded temperatures from an ESM, we draw as many realizations as we have ESM ensemble members available. For the three models we employ for validation that would be 30, 40 and 50 realizations, respectively. When forcing the emulator with emulated temperatures, we work with 100 gridded temperature fields per model and scenario, meaning we draw 100 time series from the KDE per model and scenario. We have added some information in the manuscript in L165-182 and L195-222.

*(2) How do you verify the EMU results (which is a distribution) against ESM (which is a single value)?*

AR: We verify this in multiple steps. During the fitting procedure, we compare the empirical distribution against the distribution fitted using the KDE and, for example, tune the bandwidth parameter of the KDE relying on k-fold cross-validation (L243-246). Once we have generated the full precipitation emulations (trend + variability) we compare the ensemble we generate against the entire available ESM ensemble. For example, CanESM5 has 50 ensemble members. We train our emulator on one ensemble member across SSPs. Next, we use the temperatures from the remaining 49 ensemble members as input for our emulator and pair the trend estimate from each of the ensemble members with one variability realization each, giving us 49 emulated precipitation fields. We then compare the statistics from the ESM precipitation ensemble to that of the emulated precipitation ensemble, both of which have 49 members (L208-L216 and L245-265). We focus on verifying the emulation approach as a whole, rather than verifying the individual modeling steps as we are interested in the performance of the emulation framework. The reasoning being that the GLM already captures large fractions of the precipitation signal (up to 80% of the variance of the original signal) and our focus is on validating the total variability (contribution from GLM + contribution from KDE).

*(3) What exactly does the quantile refer to in Figure 2? I assume for the ESM, the quantiles are calculated from the ensemble members (so sample size = # of ensemble member), but what about the EMU results (what are the samples size for the quantiles)?*

AR: Excellent question. We actually had not included information on emulated ensemble sizes in the manuscript. We have added information in L208-220. In short: the sample size for the ESM quantiles is always the number of ensemble members (see Table A1). The ensemble size for emulated results equals the ensemble size of the ESM for the results for the direct emulation (i.e. forcing MESMER-M-TP with temperatures from ESMs). When we emulate precipitation from emulated temperatures (Appendix C), we use 100 realizations.

*L296: Fig.3 and B1 -> should be B2?*

AR: Good catch!

*A general suggestion for the figures: it might be helpful to add subtitle like (a),(b),(c) for subplots in some figures. This can make the figure captions easier to describe and the references to the figure in the text clearer.*

AR: We agree and have adjusted where necessary.

*Figure 2 - L4: Orange/Blue lines represent precipitation estimates of a single ESM/EMU ensemble member -> should be "dashed" lines to distinguish with the solid lines*

AR: Another very good catch!

*L300-302: This is an assumption in the modeling framework, is that correct? If so, I suggest change "suggests" -> "assumes"*

AR: Adjusted accordingly.

*L314-316: It is not immediately clear to me why this suggests that the model struggles to disentangle the trend and variability?*

AR: We agree that the explanation is a bit unclear. What we meant to say is that it seems that the emulated division into trend and variability is not always accurate. A systematic underestimation of the median quantile suggests that our trend estimates are too low and the subsequent overestimation of the 10th quantile and underestimation of the 90th quantile then hints at the emulated data not having enough variability. We have adjusted the comment in the manuscript (L350-354)

We would again, like to thank referee #2 for their valuable comments. We have found them very helpful and tried to address them all as best as we could in the revised version of the manuscript.